# Simulations suggest walking with reduced propulsive force would not mitigate the energetic consequences of lower tendon stiffness

**Richard E. Pimentel[1], Gregory S. Sawicki[2,3], Jason R. Franz[1]\***

**1** Joint Department of Biomedical Engineering, UNC Chapel Hill and NC State University, Chapel Hill, North Carolina, United States of America, **2** Georgia Institute of Technology, George W. Woodruff School of Mechanical Engineering, Atlanta, Georgia, United States of America, **3** Georgia Institute of Technology, School of Biological Sciences, Atlanta, Georgia, United States of America

\* jrfranz@email.unc.edu

**Data Availability Statement:** Manuscript data and processing codes are available at the GitHub repository: https://github.com/peruvianox/kT-Fp-MetCost.

## Abstract

Aging elicits numerous effects that impact both musculoskeletal structure and walking function. Tendon stiffness ($k_T$) and push-off propulsive force ($F_P$) both impact the metabolic cost of walking and are diminished by age, yet their interaction has not been studied. We combined experimental and computational approaches to investigate whether age-related changes in function (adopting smaller $F_P$) may be adopted to mitigate the metabolic consequences arising from changes in structure (reduced $k_T$). We recruited 12 young adults and asked them to walk on a force-sensing treadmill while prompting them to change $F_P$ (±20% & ±40% of typical) using targeted biofeedback. In models driven by experimental data from each of those conditions, we altered the $k_T$ of personalized musculoskeletal models across a physiological range (2–8% strain) and simulated individual-muscle metabolic costs for each $k_T$ and $F_P$ combination. We found that $k_T$ and $F_P$ independently affect walking metabolic cost, increasing with higher $k_T$ or as participants deviated from their typical $F_P$. Our results show no evidence for an interaction between $k_T$ and $F_P$ in younger adults walking at fixed speeds. We also reveal complex individual muscle responses to the $k_T$ and $F_P$ landscape. For example, although total metabolic cost increased by 5% on average with combined reductions in $k_T$ and $F_P$, the triceps surae muscles experienced a 7% local cost reduction on average. Our simulations suggest that reducing $F_P$ during walking would not mitigate the metabolic consequences of lower $k_T$. Wearable devices and rehabilitative strategies can focus on either $k_T$ or $F_P$ to reduce age-related increases in walking metabolic cost.

## Introduction

Older adults consume energy roughly 10–30% faster than young adults to walk at the same speed or cover the same distance [1–4]. There are a number of morphological, biomechanical, neural and biochemical factors that may contribute to these higher metabolic costs. However,

**Funding:** JRF and GSS received funding for this study from National Institutes of Health (https://www.nih.gov/) Grant: R01AG058615. The funders had no role in study design, data collection and analysis, decision to publish, or preparation of the manuscript.

**Competing interests:** The authors have declared that no competing interests exist.

a recent narrative review implicated the potential interplay between age-related changes in series elastic tendon stiffness (a structural change) and push-off intensity (a functional change) during the propulsive phase of walking [5]. Model estimates suggest that lower tendon stiffness ($k_T$) yields shorter muscle fascicle lengths, decreasing the economy of force generation and increasing muscle metabolic cost [6,7]. Similarly, empirical data in younger adults show that walking with diminished push-off intensity, measured via the peak anterior/propulsive component of the ground reaction force (i.e., $F_P$), also increases the metabolic cost of walking [8]. Both of these factors (i.e., decreased $k_T$ and reduced $F_P$) are characteristic of elderly gait and have been independently studied in the context of walking economy. However, interactions between $k_T$ and $F_P$ and any resultant effects on the metabolic cost of walking have yet to be explored.

Although some discrepancies exist in the comparative literature, most human studies show that older adults exhibit lower $k_T$ and higher maximal strain during force-matched functional tasks compared to young adults [9,10]. Most reports focus on the Achilles tendon due to its relevance to walking metabolic cost and accessibility for *in vivo* imaging. Age-related decreases in Achilles $k_T$ associate with lower walking performance (shorter 6-minute walk test distance) in older adults [11,12]. This supports the role of elastic energy storage and return as a vital mechanism to minimize the metabolic cost of walking [13,14]. Altered $k_T$ has direct influence on the mechanics and economy of muscle contractions which, at least for the Achilles tendon, may influence push-off behavior. When walking at the same speed, older adults display a diminished soleus muscle operating range compared to young adults [15]. Could walking with a reduced $F_P$ mitigate the metabolic penalty we pay for reduced $k_T$?

Walking function, particularly during the push-off phase, arises from the interaction between muscle activity, muscle mechanics, and tendon elastic energy storage and return. During steady-state walking at preferred speeds, these muscle-tendon dynamics are tuned to optimize movement economy. We perform significant mechanical work during push-off–predominantly via muscle-tendon units (MTU) spanning the ankle–to propel the body forward, which exacts a metabolic cost to transition from one step to the next. Reduced $F_P$ among older adults increases walking metabolic cost by placing higher demand on more proximal leg muscles to perform mechanical work [16]. Specifically, demand for mechanical power normally accommodated by distal MTUs spanning the ankle is redistributed to more proximal MTUs spanning the hip [16,17]. This has metabolic consequences because, unlike those spanning the hip, MTUs spanning the ankle are uniquely designed for economical force production during walking with relatively shorter fascicles and longer tendons.

Although individuals can increase $k_T$ in response to mechanical stimuli from exercise [18,19], it can be challenging to quantify the role $k_T$ plays in modulating walking whole-body metabolic cost. Fortunately, musculoskeletal modeling [20–22] overcomes some of these methodological challenges. Our lab [6] and others [11,23] have used such models to reveal that decreasing $k_T$ elicits shorter muscle fiber lengths, requiring higher activations and thus higher metabolic costs to meet the task demands of walking. However, prior studies have only augmented Achilles' $k_T$ (rather than *all* of the tendons in the simulated lower limb), even though there is little experimental data to suggest that age-related decreases in $k_T$ are limited to tendons spanning the ankle. Furthermore, limiting altered $k_T$ to only the Achilles tendon may disguise other compensatory muscle actions.

Our purpose was to quantify the individual and combined effects of $k_T$ and $F_P$ on walking metabolic cost in total and at the individual-muscle level. Our central motivation was to determine whether walking with reduced $F_P$ mitigates the metabolic penalty of reduced $k_T$. However, examining larger than usual $F_P$ values is an important scientific contribution, allowing for a more comprehensive understanding of the relation between $F_P$, $k_T$, and walking metabolic cost. Understanding the full landscape of both decreasing and increasing $F_P$ in response

to biofeedback can enhance the clinical impact toward therapeutic interventions designed to enhance $F_P$. Therefore, we hypothesized that: 1) decreasing $k_T$ would contribute to higher metabolic costs during walking; and 2) $k_T$ and $F_P$ would significantly interact to affect the metabolic cost of walking. More specifically, we thought that reducing $F_P$ may offer a way to mitigate higher metabolic costs anticipated with decreased $k_T$. We also explored our experimental effects on muscle activation and fiber length to probe the mechanisms underlying the $k_T$, $F_P$, and metabolic cost landscape. Our anticipated results are intended to provide valuable insight into the tendency of older adults to walk with smaller $F_P$ and diminished ankle push-off, and how clinicians, scientists, and engineers might design devices and interventions to overcome the burden of inefficient walking.

## Methods

### Participants & experimental design

This study leverages previously published experimental data, and a detailed description of our experimental design and method can be found elsewhere [8,17]. The authors of this study did not have direct access to identifiable participant information. All participants provided written informed consent prior to engaging in any study activities. This study was approved by the University of North Carolina Institutional Review Board (IRB Protocol 18–0797). Between April and August of 2019, we recruited a convenience sample of 12 (8 female) healthy young adults (*average ± standard deviation*: age: 23.3±3.1 years; height = 1.74±0.12 m; mass = 74.7 ±14.3 kg). Participants walked 4 passes in a hallway with timing gates spaced 30 meters apart to determine their preferred walking speed.

We recorded a 5-minute, habitual-walking trial for each participant on an instrumented dual-belt treadmill (Bertec Corp., Columbus, Ohio, USA) at their preferred overground speed (1.37±0.15 m/s). We measured the peak anterior ground reaction force (i.e., propulsive force or $F_P$) from the final 2 minutes of the habitual walking trial by extracting $F_P$ from stance phases using a 20-N vertical force threshold. For our $F_P$ biofeedback, we displayed the average $F_P$ from the previous 2 steps in real time on a screen in front of the participant (Fig 1A). Alongside the real-time $F_P$, we displayed a target as a horizontal line (corresponding to the final 2-minute average $F_P$ from the habitual trial). We then familiarized each participant to our biofeedback paradigm over a 3-minute exploration trial, ensuring that each participant could readily increase and decrease $F_P$ on command prior to moving forward with the experimental protocol.

We used the average $F_P$ from the final 2 minutes of the habitual walking trial as each participant's typical $F_P$ (Norm). For experimental trials, participants walked at their preferred speed for 5-minute trials while responding to biofeedback targets of their Norm $F_P$ as well as ±20% and ±40% of Norm, presented in randomized order. In between each trial, participants rested in a seated position for at least 2 minutes.

### Musculoskeletal simulations

We performed musculoskeletal simulations in OpenSim [20–22] (version 4.1) to estimate individual muscle mechanics and metabolic energy costs. Using functional hip joint centers [24] and a static pose, we scaled all body segments of a *gait2392* model [25] for subject-specific anthropometrics in 3 dimensions. These musculoskeletal models use Hill-type muscle models (Thelen2003Muscle) with standard equilibrium equations to simulate musculotendon dynamics [26]. Model parameters based on anthropometrics (optimal fiber length and tendon slack length) were scaled using segmental scale factors. Similar to previous modeling studies [27,28], we scaled maximum isometric force by 1.5 times default value to ensure all simulations could produce the requisite joint moments. We maintained defaults for all other model parameters.

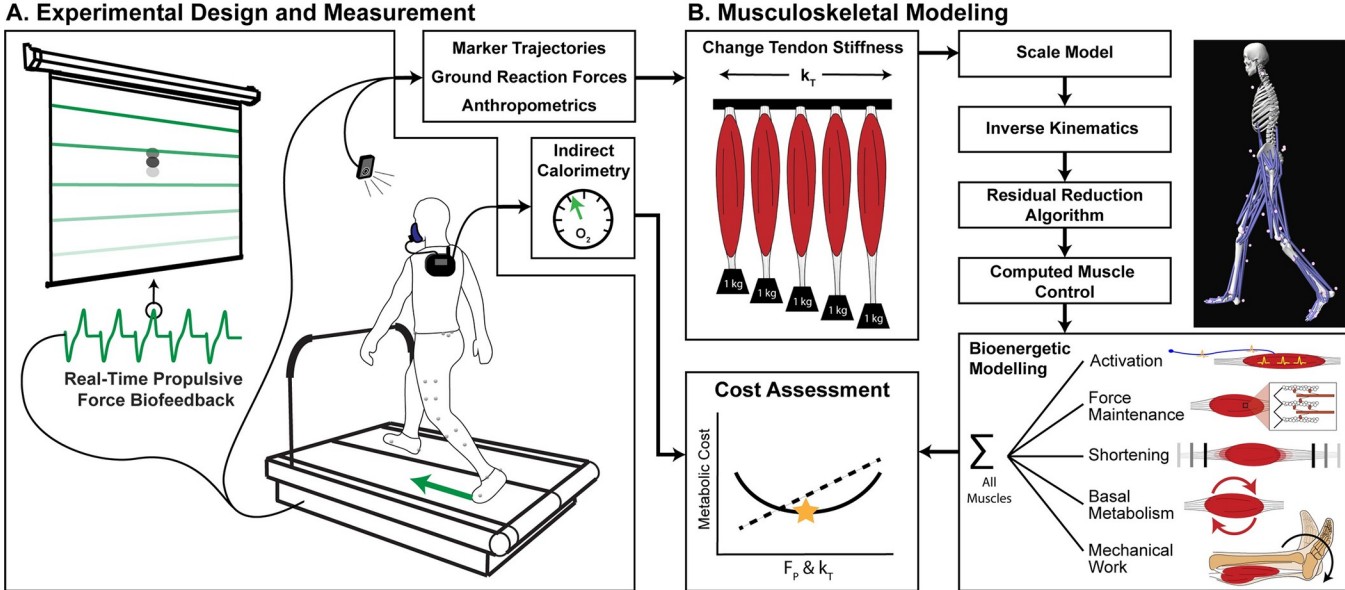

**Fig 1. Experimental design.** An overview of the (A) experimental and (B) computational methods to examine how tendon stiffness ($k_T$) and propulsive force ($F_P$) affect the metabolic cost of walking. In our experimental design (A), we asked participants to walk at their preferred speed while targeting specific $F_P$ using visual biofeedback. We simulated their movement in numerous musculoskeletal models at a range of $k_T$ levels ($\varepsilon_o$ = 2%, 3.3% (model default), 4%, 6%, and 8%) and estimate d the metabolic cost for each condition ($k_T$ & $F_P$). This figure is similar but not identical to a flowchart image we recently published [17].

We also maintained the classic optimization function that minimizes the square of muscle activations when computing muscle dynamics.

Following standard human musculoskeletal modeling techniques (Fig 1B) described previously[17], we performed computed muscle control simulations [29] across a range of tendon strain levels (i.e., $\varepsilon_o$, tendon strain at maximum isometric force). Specifically, we simulated $\varepsilon_o$ at 2%, 3.3% (model default), 4%, 6%, and 8% for all model tendons (92 in total). We changed tendon strain because we could not directly alter tendon stiffness in the musculoskeletal model. We chose these strain values because they lie within previous simulation studies [6,30], contain the expected range of tendon strain for younger and older adults [11,31,32], and, in the case of 2% $\varepsilon_o$, provides a stiffer comparison versus default (3.3% $\varepsilon_o$).

During the simulations, we probed muscle metabolic costs using the Bhargava 2004 and Umberger 2010 models [33,34], which are readily available in OpenSim. For a conservative estimate, we report the average of these two bioenergetic models. All metabolic costs presented in this study result from the musculoskeletal models, not from indirect calorimetry. Furthermore, we refer to muscle names using the standard nomenclature from the *gait2392* model MTU actuators. Finally, we follow syntax from the metabolic models and report the sum of all modeled muscles as *Total* metabolic cost.

To follow common terminology, we use the term tendon stiffness ($k_T$) generally throughout, rather than tendon strain ($\varepsilon_o$). We acknowledge that this complicates the narrative due to the inverse relation between stiffness and strain (2% $\varepsilon_o$ = most stiff, 8% $\varepsilon_o$ = least stiff). We clarify these parameters in our figures by labeling both strain ($\varepsilon_o$) and stiffness ($k_T$) whenever possible.

### Data reduction & statistics

From a selection of gait cycles over the final 2 minutes of each trial, we reduced simulation input data (motions & forces) down to one gait cycle on each side by selecting the first left and

right stride from the 10-second window with the most accurate biofeedback targeting during the two minutes (as performed previously [17]). We performed a total of 600 computational simulations (i.e., 12 subjects, 5 biofeedback targets, 5 $\varepsilon_o$ values, 2 strides *[left & right]*). We averaged metabolic cost estimates bilaterally for each condition prior to statistical analysis. Outcome variables included total and individual-muscle metabolic costs, reported on average and as a percentage of the gait cycle. We simplified muscles with multiple lines of action (i.e., glut_med1, glut_med2, glut_med3) by summing each component for the whole muscle (i.e., glut_med).

To analyze the effects on stride-average walking metabolic cost, we performed two-way repeated measure analyses of variance (ANOVA) to test for main effects of and interactions between $k_T$ and $F_P$ at whole-body and individual-muscle levels ($\alpha = 0.05$). Alongside the ANOVA results, we also report partial eta squared ($\eta_p^2$) effect sizes. Similarly, to assess the effects on walking metabolic cost as a percentage of the gait cycle, we used statistical parametric mapping [35,36] to quantify main effects of $k_T$ and $F_P$ ($\alpha = 0.001$). We also performed Pearson correlations to explore associations between primary variables (metabolic cost, $k_T$, and $F_P$) and muscle-level determinants (i.e., activation and fiber length). We performed all statistical calculations using the Pingouin and SciPy packages [37,38]. For transparency and to support open science, we provide our simulation data and processing scripts at: https://github.com/peruvianox/kT-Fp-MetCost.

## Results

### Total metabolic cost

We found significant main effects of $k_T$ (p<0.001, $\eta_p^2 = 0.423$, Fig 2 horizontal axis) and $F_P$ (p = 0.014, $\eta_p^2 = 0.244$, Fig 2 vertical axis) on total metabolic cost. We did not find a significant interaction between $k_T$ and $F_P$ (p = 0.162, $\eta_p^2 = 0.111$). In general, total metabolic cost increased as $k_T$ decreased ($\varepsilon_o$ increased) or as $F_P$ deviated from the Norm intensity.

When viewed across the gait cycle (Fig 3), we found that $k_T$ and $F_P$ affect instantaneous metabolic cost differently across various phases of the gait cycle. We found significant main effects of $k_T$ ($\varepsilon_o$) during early stance (10–16% gait cycle), late stance (48–52% and 55–60% gait cycle) and late swing (92–100% gait cycle). Alternatively, we found significant main effects of $F_P$ during mid-to-late stance (~25–32% and 40–45% gait cycle) and mid-to-terminal swing (72–80% and 97–100% gait cycle). In general, the effects of $k_T$ and $F_P$ on total metabolic cost occurred at different times of the gait cycle, not simultaneously. Although we found no interaction between $k_T$ and $F_P$ for total metabolic cost on average, we observed a few interactions during early stance and push-off (Fig 3).

### Individual muscle metabolic costs

Table 1 shows the metabolic costs for all modeled muscles, ranked in order of energy consumption, and how their consumption varied as a function of $k_T$ and $F_P$. Ten different muscles contributed >4% (>2% unilateral) to total metabolic cost. The three highest energy-consuming muscles (*glut_med*, *rec_fem*, *and soleus*) each contributed >8% (>4% unilateral).

Fig 4 graphically summarizes the effects of $k_T$ and $F_P$ for the top 12 most costly muscles, which accounted for 70.8% (35.4% unilateral) of the total metabolic cost at default $k_T$ and Norm $F_P$. Eight of top 12 individual-muscle contributors to total metabolic cost showed a main effect of $k_T$ (horizontal arrows), including all three muscles spanning the ankle and three out of the five muscles spanning the knee. Nine of the top 12 contributors to total metabolic cost showed a main effect of $F_P$ (vertical arrows), including six of the seven muscles spanning

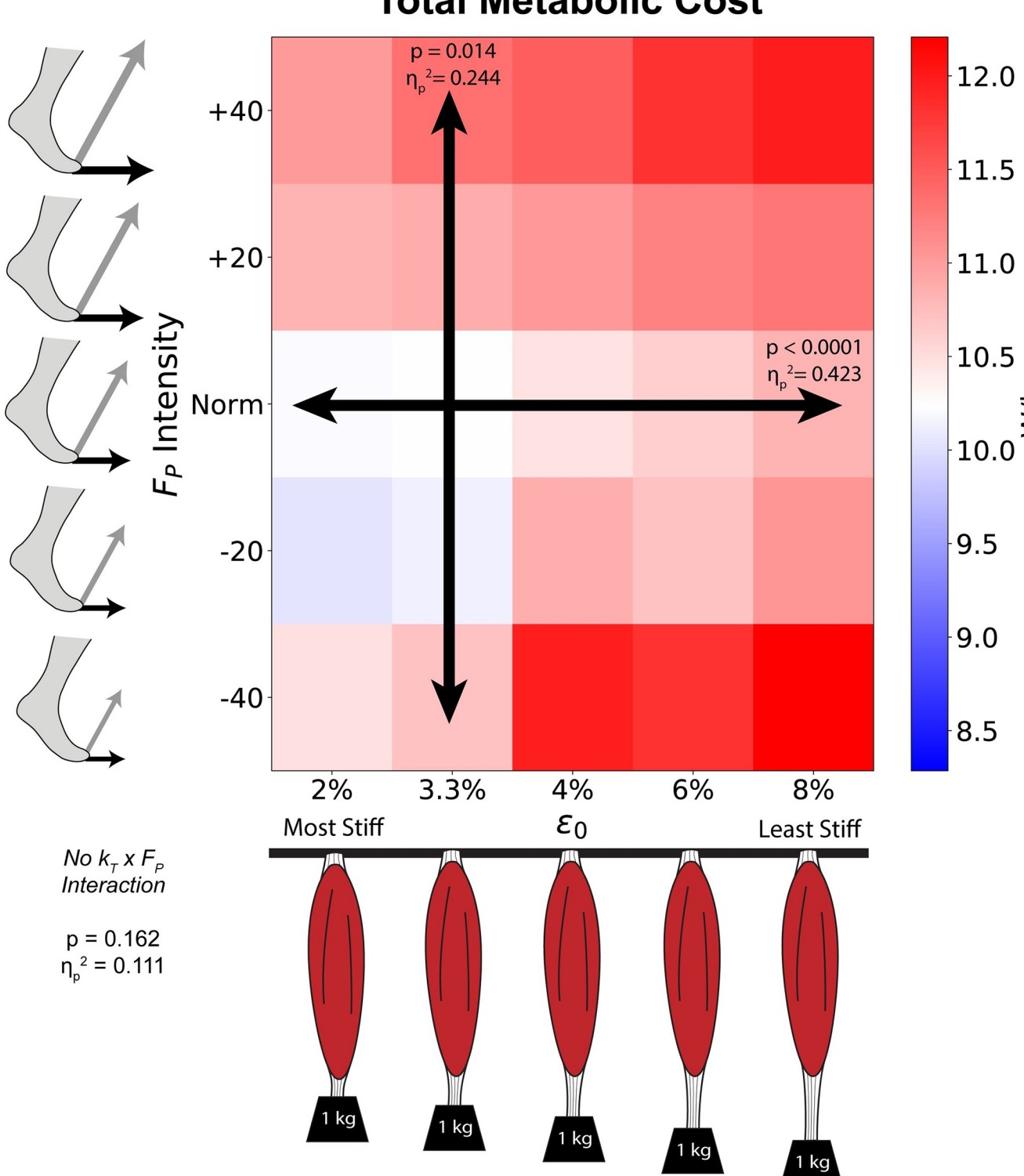

**Fig 2. Total average metabolic costs.** We show how average total metabolic cost varies across $k_T$ (horizontal axis) and $F_P$ intensity (vertical axis). This heatmap is color coded for the reference metabolic cost (default $k_T$ and Norm $F_P$ intensity) to be displayed in white, with higher costs in red and lower costs in blue. We found significant ANOVA main effects separately for $k_T$ (horizontal arrow) and $F_P$ (vertical arrow), but no interaction (no diagonal arrow) between them.

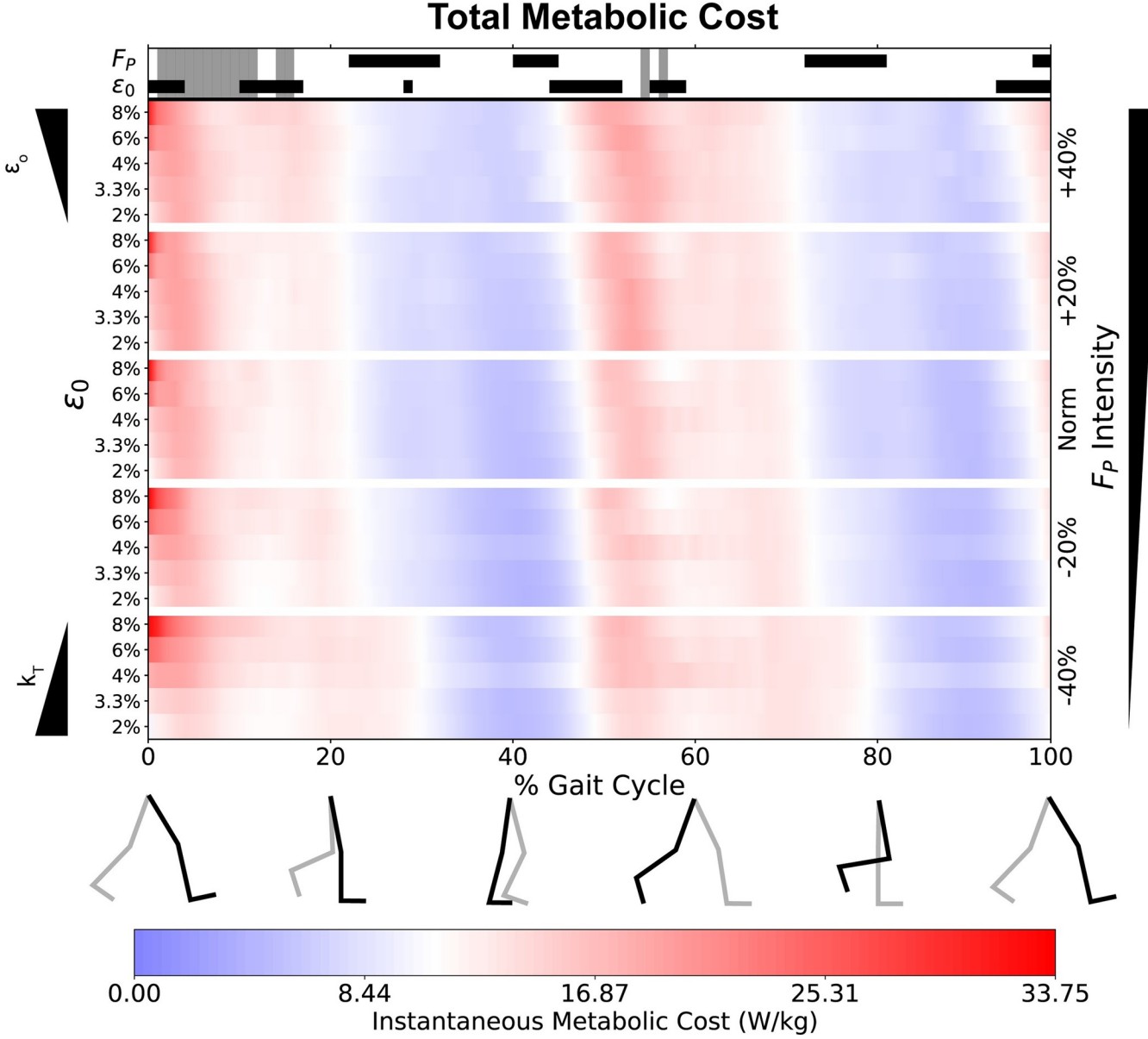

**Fig 3. Total instantaneous metabolic costs.** We show instantaneous, whole-body metabolic cost as a percentage of the gait cycle, across both $k_T$ (left minor axes) and $F_P$ (right major axis). The average metabolic cost at default $k_T$ and Norm $F_P$ is normalized to white. We display higher costs in red, and lower costs in blue, with the color intensity profiles even between the two. At the top of the figure, we show periods with significant main effects for $k_T$ (or $\varepsilon_o$) and $F_P$. The black horizontal bars indicate a significant repeated measures ANOVA main effect via statistical parameter mapping.

the hip (all but *rec_fem*). Finally, five of the top 12 most energy-consuming muscles showed an interaction effect (diagonal arrows), including all three muscles spanning the ankle.

Fig 5 shows how the individual muscle metabolic costs vary across the gait cycle. We see interaction effects at the individual-muscle level during specific instances of the gait cycle (gray shaded regions along the top bar). For example, on average, we found significant interactions between $k_T$ and $F_P$ for *soleus*, *glut_max*, *tib_ant*, and *med_gas* (Table 1). We can see these interactions during early-to-mid stance for *glut_max* (Fig 5C); early stance, push-off, and late swing for *soleus* (Fig 5J); intermittently throughout stance phase for *med_gas* (Fig 5K); and push-off for *tib_ant* (Fig 5L).

**Table 1. Muscle metabolic costs.**

| Rank | Muscle | Metabolic Cost at Default | | $F_P$ | | $k_T$ | | Interaction | |
|---|---|---|---|---|---|---|---|---|---|
| | | W/kg | % | $p$ | $\eta_p^2$ | $p$ | $\eta_p^2$ | $p$ | $\eta_p^2$ |
| - | Total | 10.22 | 100.0 | **0.014** | **0.244** | **0.000** | **0.423** | 0.162 | 0.111 |
| 1 | **glut_med** | 0.46 | 4.5 | **<0.001** | **0.467** | 0.556 | 0.065 | 0.227 | 0.103 |
| 2 | **rect_fem** | 0.46 | 4.5 | 0.671 | 0.051 | 0.607 | 0.059 | 0.867 | 0.053 |
| 3 | **soleus** | 0.44 | 4.3 | 0.105 | 0.157 | **<0.001** | **0.534** | **0.015** | **0.154** |
| 4 | **glut_max** | 0.37 | 3.6 | **<0.001** | **0.528** | **<0.001** | **0.569** | **0.001** | **0.191** |
| 5 | **psoas** | 0.34 | 3.3 | **0.023** | **0.223** | **0.027** | **0.216** | 0.410 | 0.087 |
| 6 | ercspn* | 0.33 | 3.2 | 0.641 | 0.054 | 0.080 | 0.169 | 0.673 | 0.069 |
| 7 | **iliacus** | 0.30 | 3.0 | **0.026** | **0.218** | 0.175 | 0.132 | 0.414 | 0.087 |
| 8 | **bifemsh** | 0.29 | 2.8 | **0.000** | **0.394** | 0.093 | 0.162 | 0.486 | 0.081 |
| 9 | **tib_ant** | 0.28 | 2.8 | 0.081 | 0.168 | **<0.001** | **0.509** | **0.036** | **0.140** |
| 10 | **med_gas** | 0.26 | 2.5 | **0.001** | **0.345** | **<0.001** | **0.538** | **0.004** | **0.174** |
| 11 | **semimem** | 0.18 | 1.7 | **0.002** | **0.311** | **<0.001** | **0.643** | 0.653 | 0.070 |
| 12 | **vas_lat** | 0.13 | 1.3 | **0.000** | **0.406** | **<0.001** | **0.412** | **0.032** | **0.142** |
| 13 | **bifemlh** | 0.11 | 1.1 | **0.000** | **0.481** | **<0.001** | **0.760** | 0.263 | 0.099 |
| 14 | sar | 0.11 | 1.1 | 0.555 | 0.065 | 0.743 | 0.043 | 0.485 | 0.081 |
| 15 | intobl | 0.10 | 1.0 | 0.871 | 0.027 | 0.355 | 0.093 | 0.510 | 0.080 |
| 16 | ext_dig | 0.10 | 1.0 | **0.017** | **0.236** | **<0.001** | **0.529** | 0.066 | 0.129 |
| 17 | vas_int | 0.07 | 0.7 | **<0.001** | **0.440** | **0.001** | **0.334** | **0.016** | **0.153** |
| 18 | lat_gas | 0.07 | 0.7 | **<0.001** | **0.446** | **0.001** | **0.343** | 0.190 | 0.107 |
| 19 | vas_med | 0.07 | 0.7 | **<0.001** | **0.433** | **0.002** | **0.315** | **0.004** | **0.173** |
| 20 | add_long | 0.07 | 0.7 | 0.689 | 0.049 | **0.024** | **0.221** | 0.448 | 0.084 |
| 21 | glut_min | 0.06 | 0.60 | **<0.001** | **0.552** | 0.622 | 0.057 | 0.359 | 0.091 |
| 22 | extobl | 0.06 | 0.60 | **0.018** | **0.233** | **<0.001** | **0.557** | 0.146 | 0.113 |
| 23 | add_mag | 0.06 | 0.60 | **<0.001** | **0.646** | **<0.001** | **0.455** | 0.243 | 0.101 |
| 24 | semiten | 0.05 | 0.50 | 0.060 | 0.183 | **<0.001** | **0.434** | 0.055 | 0.132 |
| 25 | tib_post | 0.03 | 0.30 | **<0.001** | **0.472** | **<0.001** | **0.736** | 0.437 | 0.085 |
| 26 | tfl | 0.03 | 0.30 | 0.058 | 0.184 | **<0.001** | **0.603** | 0.586 | 0.075 |
| 27 | per_long | 0.03 | 0.30 | **<0.001** | **0.403** | **<0.001** | **0.567** | 0.190 | 0.107 |
| 28 | quad_fem | 0.02 | 0.20 | 0.701 | 0.047 | 0.167 | 0.134 | 0.365 | 0.090 |
| 29 | peri | 0.02 | 0.20 | 0.224 | 0.119 | **<0.001** | **0.496** | 0.237 | 0.102 |
| 30 | add_brev | 0.02 | 0.20 | 0.832 | 0.032 | 0.532 | 0.068 | 0.360 | 0.091 |
| 31 | grac | 0.02 | 0.20 | **0.035** | **0.206** | **0.013** | **0.246** | 0.186 | 0.108 |
| 32 | ext_hal | 0.02 | 0.20 | **0.005** | **0.281** | **<0.001** | **0.502** | 0.150 | 0.112 |
| 33 | pect | 0.01 | 0.10 | 0.482 | 0.074 | 0.541 | 0.067 | 0.553 | 0.077 |
| 34 | per_tert | 0.01 | 0.10 | **0.004** | **0.290** | **0.002** | **0.313** | 0.274 | 0.098 |
| 35 | per_brev | 0.01 | 0.10 | 0.152 | 0.139 | **0.031** | **0.211** | 0.654 | 0.070 |
| 36 | flex_hal | 0.01 | 0.10 | **0.005** | **0.285** | **<0.001** | **0.761** | 0.091 | 0.123 |
| 37 | flex_dig | 0.01 | 0.10 | **0.030** | **0.212** | **<0.001** | **0.868** | 0.346 | 0.092 |
| 38 | gem | 0.00 | 0.00 | 0.947 | 0.016 | **<0.001** | **0.437** | 0.384 | 0.089 |

We rank the top 20 individual muscles that contribute to whole-body metabolic cost of walking in the default setting ($F_P$ = Norm and $\varepsilon_o$ = 3.3%). We display each muscle's average requirement for net metabolic power in absolute (W/kg) and relative (%) terms. In addition, we demonstrate how these individual muscles respond to changes in $k_T$, $F_P$, and interaction by reporting the ANOVA main effect (p value) and effect size ($\eta_p^2$). All muscle metabolic costs are unilateral, with the total shown as a bilateral sum. Muscle names follow conventional nomenclature from the Gait2392 model developed by OpenSim. Bold muscle names indicate the top 12 contributors to metabolic cost. *Note: although *ercspn* (erector spinae) contributed to 3.2% of the metabolic cost, placing it as the 6[th] highest individual muscle, it is not a lower-body muscle and its cost did not change with $k_T$ or $F_P$. Thus, we have omitted *ercspn* from further analysis (in the figures and discussion).

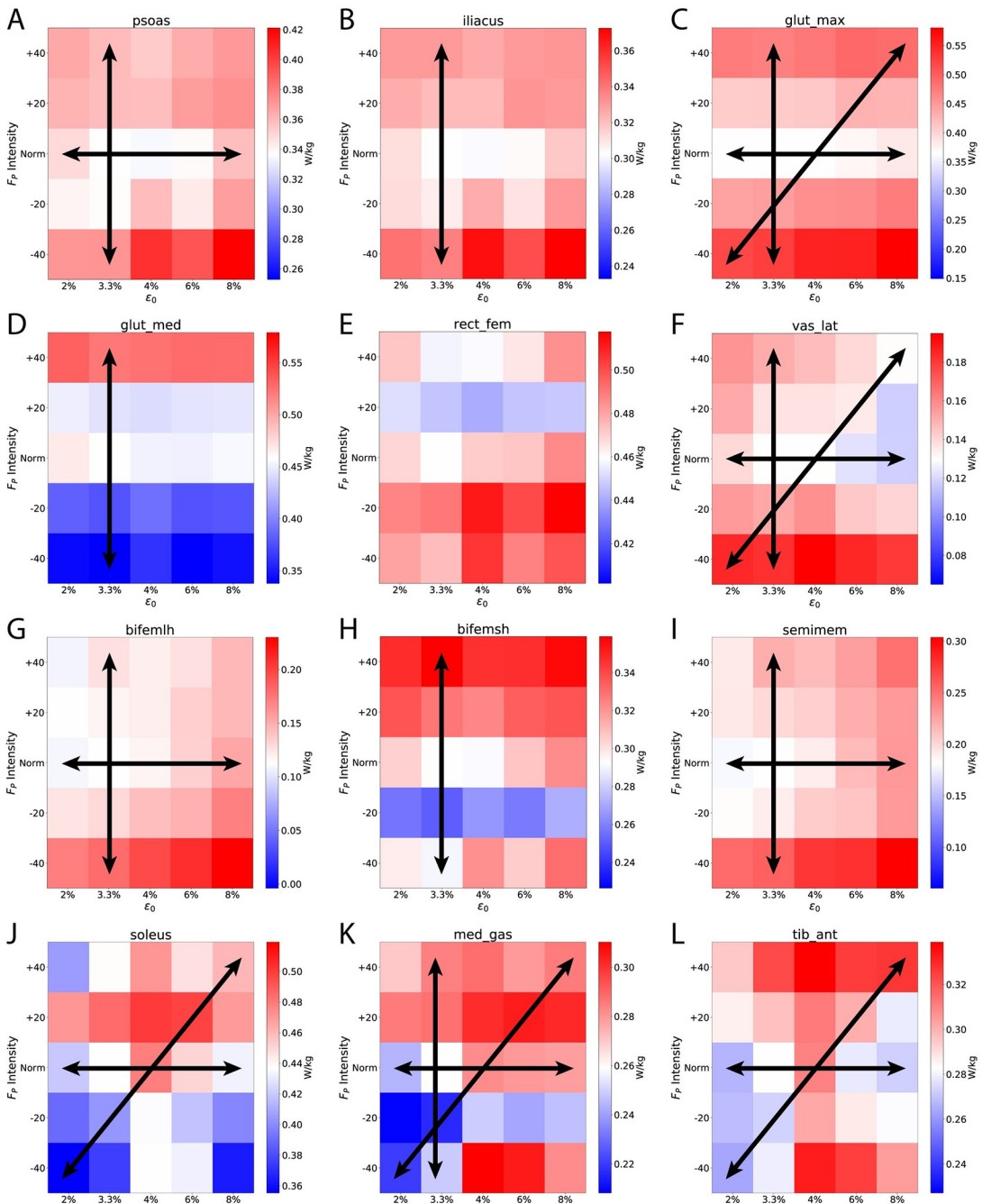

**Fig 4. Individual muscle average metabolic costs.** Individual muscle metabolic costs respond uniquely across $k_T$ and $F_P$. In this figure, we show the top 12 lower-body muscles that contribute to walking metabolic cost (Table 1). We oriented the heatmaps with proximal musculature (hip) towards the top, and distal musculature (ankle) towards the bottom. Like Fig 2, we normalized each heatmap for the default $k_T$ and $F_P$ values (3.3% and Norm, respectively) to be shown in white, with higher costs in red and lower costs in blue. Within each heatmap, we show significant ANOVA main effects via horizontal, vertical, and diagonal arrows indicating significant effects for $k_T$, and $F_P$, and interaction, respectively.

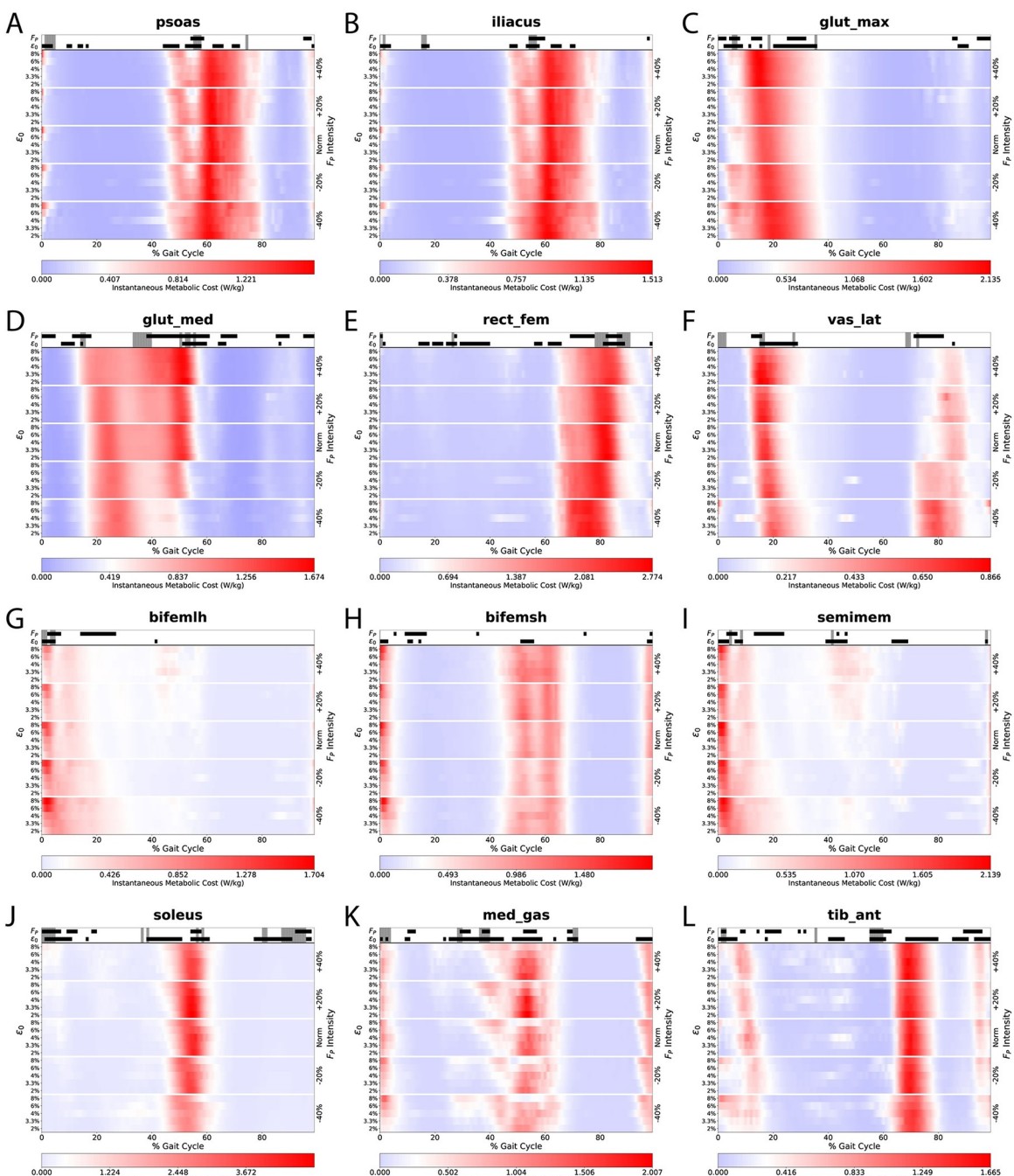

**Fig 5. Individual muscle instantaneous metabolic costs.** Timing and intensity of individual-muscle metabolic costs change when varying $k_T$ and $F_P$. In this figure, we show 12 lower-body muscles as in Fig 4, now including instantaneous metabolic cost across the gait cycle. These heatmaps are designed similar to the whole-body costs in Fig 3, with $k_T$ on the left minor vertical axis, $F_P$ on the right vertical major axis, and relative time (% GC) on the horizontal axis. We show significant ANOVA main effects from instantaneous statistical parametric mapping using blocks ($\varepsilon_o$ and $F_P$) and shaded regions (interactions) along the top bar.

## Individual muscle fiber lengths and activation levels

We found a significant association between metabolic cost and mean activation (Fig 6A) but not for normalized mean fiber length (Fig 6B). $F_P$ had a mixed influence on the relationship between activation and fiber length (Fig 6C), whereas $k_T$ (strain) showed a strong tendency for

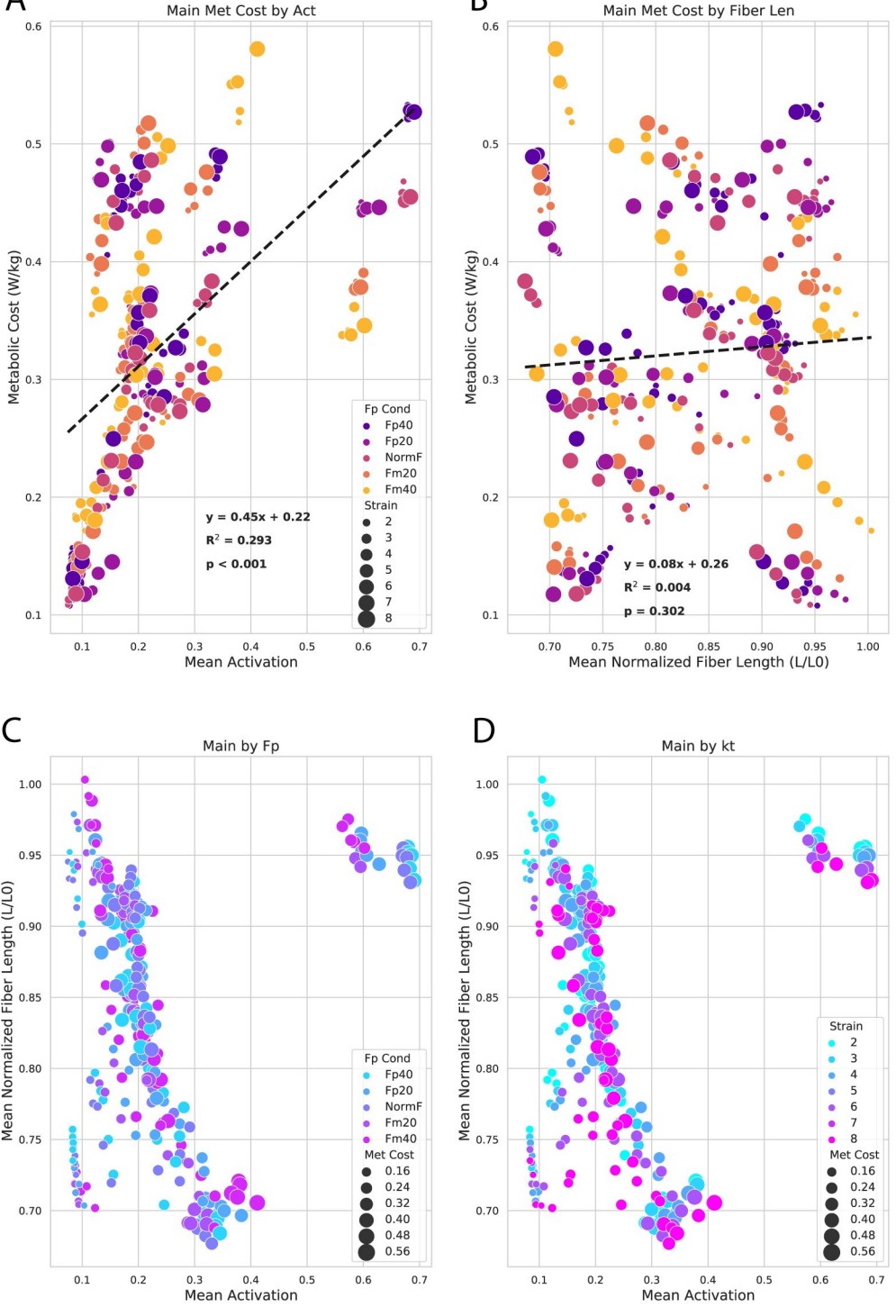

**Fig 6. Activation, fiber length, and metabolic costs.** These scatterplots show general associations between the underlying muscle-tendon dynamics of activation and fiber length and their influence on the relationships between metabolic cost, $F_P$, and $k_T$. Each circle represents the subject-average outcome at a given activation, fiber length, metabolic cost, $F_P$, and $k_T$ for the top dozen individual-muscle contributors to metabolic cost (also shown in Figs 4 and 5). We found a significant association between metabolic cost and mean activation (A) but not for normalized mean fiber length (B). $F_P$ (C) seemed to have more mixed influence on the relationship between activation and fiber length, whereas $k_T$ (strain, D) showed a strong effect for shorter fiber lengths and higher activations (downward & rightward shift from blue (most stiff) to pink (least stiff)).

shorter fiber lengths and higher activations (Fig 6D, downward & rightward shift from blue [most stiff] to pink [least stiff]).

For additional context and transparency behind our simulations, we provide supplementary figures showing the simulated activation levels (S1 and S3 Figs) and fiber lengths (S2 and S4 Figs) for total and individual-muscle metabolic costs. We also show the average activation levels and normalized fiber lengths at default $k_T$ and Norm $F_P$ for these muscles in S1 and S2 Tables. On average across the gait cycle, the top dozen energy-consuming muscles (bolded muscle names) tended to have higher activation levels (higher ranks in S1 Table) and shorter fiber lengths (lower ranks in S2 Table) compared with the rest of the musculature.

We conclude our results by providing additional data on the triceps surae musculature (S5 Fig) and on lower-body sagittal plane kinematics across the biofeedback conditions (S6 Fig). S5 Fig summarizes comparisons between the three triceps surae muscles, as *lat_gas* was not a member of the top 12 contributors to metabolic cost. The cost landscape for *lat_gas* largely aligned with that of *med_gas* but with smaller metabolic costs. In terms of kinematics, we found that reducing $F_P$ tended to decrease sagittal plane hip, knee, and ankle joint ranges of motion (S6 Fig). Conversely, increasing $F_P$ tended to increase ankle extension near push-off (~60% gait cycle).

## Discussion

A hybrid of experimental and computational approaches allowed us to investigate whether age-related changes in function (adopting smaller $F_P$) affected the metabolic consequences arising from those in structure (having lower $k_T$). This combined experiment yielded simulated instantaneous metabolic costs, providing the ability to identify muscle level responses across two of the predominant aging-related factors that contribute to increased walking metabolic cost.

Our data support our first hypothesis, that decreasing $k_T$ increases the metabolic cost of walking for total costs as well as nearly all individual muscles. Our second hypothesis was more nuanced. We reject our second hypothesis at the whole-body level, as total metabolic cost only increased as $k_T$ and $F_P$ decreased (rightward and downward shift in Fig 2). However, certain muscles did display cost savings when emulating age-related changes (lower $k_T$ and reduced $F_P$). In particular, the triceps surae muscles are a primary determinant of $F_P$ generation and exhibited a rightward and downward shift (S5A–S5C Fig). For example, if we compare the default condition (Norm $F_P$ and 3.3% $\varepsilon_o$) with a reasonable aging shift (i.e., -20% $F_P$ and 6% $\varepsilon_o$) we see a 4.6% increase in total metabolic cost (Fig 2) as well as a 7% cost reduction for the triceps surae (4.4%, 6.8%, and 9.9% reductions for the *soleus*, *med_gas*, and *lat_gas*, respectively, Figs 4 & S5). Thus, these simulations suggest that older adults may adopt a strategy with local reductions in metabolic cost, but with increased costs when summed at the whole-body level.

We found no interaction at the whole-body level between $k_T$ and $F_P$ on metabolic cost using simulations of walking at a constant speed in young adults. Rather, $k_T$ and $F_P$ each independently affect walking metabolic cost, with no evidence that walking with smaller $F_P$ may be adopted to mitigate the metabolic consequences arising from reduced $k_T$.

By exploring individual muscle contributions to total metabolic cost, we show unique intermuscular responses to alterations in $k_T$ and $F_P$. In the following sections, we further interpret these cumulative results from young adults walking across a range of $F_P$ intensities in the context of characteristic changes in structure and gait function in older adults. These findings highlight the importance of structural components ($k_T$) and functional behavior ($F_P$) as determinants of walking economy.

### Effects of simulated changes in tendon stiffness

It is well documented that $k_T$ decreases with age and physical inactivity. Lower $k_T$ associates with worse walking performance [11,12] generally via two mechanisms: directly via changes in tendon elastic energy storage, return, and MTU power generation, and indirectly by compelling shorter muscle lengths. For example, altering gastrocnemius and solus reference tendon strain *in silico* from 3–11% yields vastly different tendon and muscle lengths during stance, and thus the timing and magnitudes of tendon and muscle power [6]. Our main premise is that operating a less-stiff tendon elicits shorter muscle lengths to compensate for greater tendon elongation for a given force output and MTU length [6]. These effects presume higher excitations at a metabolic penalty or may sufficiently alter musculotendon dynamics to alter muscle activation timing.

In a similar study to ours, increasing Achilles tendon stiffness did not significantly decrease walking energy cost in older adults [23]. Biological ranges of $k_T$ incorporate the local cost minima of walking [23,39]. We generally interpret reduced $k_T$ as pathophysiological change that would benefit from intervention. However, as a potential alternative interpretation, older adults likely participate in less physical activity, and thus their lower stiffness may be a physiological adaptation to optimize metabolic cost for their daily activities [10].

Altering only the Achilles $k_T$ in older and younger adults yields relatively small effects (~1.5% change from baseline) on whole-body metabolic cost across a range of walking speeds [23]. Although the Achilles tendon and triceps surae musculature may be most impacted by changing $k_T$, we cannot assume that only the Achilles tendon would be affected by altered $k_T$. Thus, we allowed for uniform changes in $k_T$ across all MTUs in our musculoskeletal models. In doing so, we found a rather large effect size for $k_T$ ($\eta_p^2 = 0.423$, explaining ~42% of the variance in metabolic cost) [40]. When considering Norm $F_P$ alone, reducing $k_T$ (increasing $\varepsilon_o$ up to 8%) yielded a 5% increase in metabolic cost on average (10.2 W/kg at 3.3% $\varepsilon_o$ vs. 10.8 W/kg at 8% $\varepsilon_o$). Our findings agree well with the literature that reducing $k_T$ increases the metabolic cost of walking [6,30].

We can see instantaneous effects of $k_T$ on total metabolic cost particularly during the beginning of the push-off phase (i.e., around 50% of the gait cycle, Fig 3). At that instant, simulations with lower $k_T$ (i.e., 6–8% $\varepsilon_o$) showed an earlier increase in metabolic cost than those with higher $k_T$ (i.e., 2–4% $\varepsilon_o$). Breaking this down to individual muscles, this $k_T$ effect on metabolic cost arises from ankle extensors (i.e., *soleus* and *med_gas*) and hip flexors (i.e., *psoas* and *iliacus*). We also saw main effects of $k_T$ on activation for *med_gas* (S3K Fig) and on fiber length for all four muscles (S4A, S4B, S4J and S4K Fig). Operating against their least stiff tendons, the *soleus* and *med_gas* showed large metabolic effects of $k_T$, functioned at shorter fiber lengths, and did not lengthen much during mid-stance phase (10–50% gait cycle). Conversely, operating against their stiffest tendons, these same muscles exhibit lengthening behavior during mid stance, and function at longer fiber lengths during push off. If timed appropriately, tendons with lower stiffness may exploit tendon elongation to store spring potential energy for powering forceful activities (such as $F_P$) later. Indeed, *med_gas* MTUs with the less-stiff tendons exhibit bursts of higher costs at the beginning of push-off (~40% gait cycle), but dramatic cost savings later in push-off (50–60% gait cycle, Fig 5K). For a clearer distinction of these fiber length differences, please refer to S7 Fig.

Qualitatively looking at muscle dynamics, $k_T$ also seemed to show a direct impact on mean activation and fiber length demonstrating a clear shift towards shorter fiber lengths and higher activations (Fig 6D). Our walking simulations and metabolic outcomes are consistent with the majority of outcomes from other studies, finding that individual muscle actions and metabolic cost during walking are highly sensitive to changes in $k_T$ [18,23,39,41].

The large effects of $k_T$ on metabolic cost may arise because it impacts not only the force-length, but also the force-velocity relation of muscle. For force-length, a less-stiff tendon would compel shorter muscle operating ranges when MTU length is constrained (as it was in this study). Shorter muscles generate less force, thus requiring a higher activation at a given submaximal force output, yielding higher metabolic costs. For force-velocity, a less-stiff tendon would elongate more when transmitting a specific force. In MTU-shortening activity, this would require the muscle to shorten more rapidly, requiring higher activations and thus, higher costs.

### Effects of changing propulsive force

In this study, we used $F_P$ as our proxy for functional changes, due to its strong association with walking speed [42,43] and hallmark decline among older adults [44–47]. We recently discovered empirical evidence that a diminished $F_P$ increases metabolic cost, explained via the distal-to-proximal redistribution of muscle workload [16]. Not surprisingly, walking with larger $F_P$ at a fixed speed, at least among younger adults, also increases measured walking metabolic cost. Our whole-body bioenergetic predictions support these earlier measurements.

Interestingly, we did not an effect of $F_P$ on total metabolic cost during push-off phase (50–60% gait cycle, Fig 3). Similar studies that increased/decreased propulsion found increases in neuromuscular drive to the ankle extensors [48,49]. In agreement with the literature, we do see individual muscle effects (for metabolic cost, activation, and fiber length) during this phase for the hip flexors (*psoas* and *ilicus*), hip abductor (*glut_med*), and ankle extensors (*soleus* and *med_gas*). These individual muscle outcomes reveal the compensatory costs of walking with altered $F_P$. For example, walking with greater $F_P$ exacts higher ankle extensor metabolic costs. Conversely, walking with a diminished $F_P$ requires the hip flexors to compensate with higher metabolic costs to drive hip flexion. As a particularly interesting outcome, *glut_med* (a hip abductor), operated at higher costs when walking with larger $F_P$ (Figs 4D and 5D). We suggest this may relate to an increased need for hip stability while transmitting larger forces from the ankle extensors to the body's center of mass [50].

Our protocol prompted changes in $F_P$ using targeted biofeedback which changed lower body kinematics (S6 Fig). Because even minor changes in joint kinematics can influence tendon length (and therefore strain) [51], we would infer the inverse is also true (where minor changes in tendon strain can influence joint kinematics). This inverse relationship is supported in part by differing joint kinematics between younger and older adults during walking [52]. Because there is no way to control for all experimental variables ($k_T$, $F_P$, simulated metabolic cost, and effects of aging) in this study, we note that joint kinematics did change in response to $F_P$, but were constrained across the $k_T$ simulations.

### Interactions between propulsive force and tendon stiffness

Although we did not find an interaction between $k_T$ and $F_P$ for total metabolic cost on average (Fig 2), we did see periods of significant interactions across the gait cycle (Fig 3) occurring during early stance (~5–20%) and push off (~55%). Because the instantaneous total metabolic costs in Fig 3 are comprised of bilateral data, the periods of significant interaction (5–10% and 55–60% gait cycle) may arise from the bilateral nature of gait, with both distal and proximal muscles contributing to the interactions across both time periods. As total metabolic cost is the sum from individual muscles, we discuss the individual muscles likely to explain these interactions.

Of the 5 muscles with a significant $k_T$ and $F_P$ interaction on average (Fig 4), *glut_max* and *vas_lat* likely explain those during early stance. The cost profile for *glut_max* during early

stance (5–10%) exhibited higher metabolic costs for both lower $F_P$ (major vertical axis) and lower $k_T$ (minor vertical axis, Fig 5C). The higher costs due to lower $F_P$ aligns with the well-documented redistribution of muscle work from the ankle to the hip [16]. The effect of $k_T$ likely arises from the relative fiber length differences imposed by tendon constraints. During this concentric activity of *glut_max* (S4C Fig), some of the MTU shortening is lost via tendon elongation, requiring *glut_max* to contract more to overcome the tendon lengthening, and thus requiring higher costs (Fig 5C).

In an eccentric example, *vas_lat* consumes more energy with higher $k_T$ (Fig 5F) during eccentric activity during early-to-mid stance (15–20% gait cycle, S3F and S4F Figs). *Vas_lat* muscle fibers attached to less-stiff tendons (higher $\varepsilon_o$) may maintain relatively similar lengths as the compliant tendons uptake changes in length. Whereas, when attached to stiff tendons, *vas_lat* muscle fibers must elongate to account for the length change (see white/green areas in the minor axis of S4F Fig at 15–20% gait cycle, particularly visible for Norm and +40% $F_P$ conditions). This eccentric activity is more costly than isometric activity, exhibiting one aspect of the $k_T$−$F_P$ interaction.

Of the final 3 muscles that experienced significant interactions on average (Fig 4J–4L), *soleus* and *tib_ant* demonstrated instantaneous interactions near the end of push-off (55–60% gait cycle, Fig 5J and 5L). The *soleus* showed lower costs with reduced $F_P$ as well as lower costs with lower $k_T$ (higher $\varepsilon_o$). The lower metabolic costs with reduced $F_P$ is straightforward and aligns with lower soleus activation (S3J Fig). The lower metabolic cost at lower $k_T$ likely arises from elastic recoil of the Achille's tendon providing some MTU-shortening this late into push-off. The interaction between $k_T$ and $F_P$ for the soleus also aligns with interaction timing for total metabolic cost (Fig 3). Interestingly, *med_gas* did not share the instantaneous $k_T$ and $F_P$ interaction with the soleus during push-off, even though there were independent significant effects for $k_T$ and $F_P$ separately (Fig 5).

## Relevance to precision rehabilitation

The triceps surae musculature (*soleus*, *med_gas*, and *lat_gas*) provides a significant portion of the work and power for gait by generating $F_P$. Neuromuscular adaptations conserve energy on an individual-muscle or muscle-group level. We found that neuromuscular responses to minimize activation (i.e., reducing $F_P$ in response to biofeedback) may successfully mitigate costs for the triceps surae muscles (S5 Fig), but do not necessarily reduce total energy costs (Fig 2) due to compensatory neuromuscular responses in other muscles (Fig 4).

Our results show that $k_T$ and $F_P$ do not interact in their effects on the metabolic cost of walking–at least in young adults. While this is a novel finding, it is most relevant when placed in the context of walking among older adults. Specifically, our data suggest that smaller observed Fp in older adults would not be an effective mitigation strategy to conserve total energy costs energy to counteract lower $k_T$. A logical extension of that interpretation is that interventions designed to reduce the metabolic cost of walking in older adults are not subject to a trade-off and can independently address either or both reduced $k_T$ and diminished $F_P$.

We envision several viable options for solutions that could address the metabolic consequences of reduced $k_T$ and diminished $F_P$. For example, plyometric training may help increase $k_T$ [19,53] and biofeedback-based gait retraining may help improve $F_P$ [54]. Thus far, training strategies to reduce the metabolic cost of walking among older adults have been generally unsuccessful, primarily because they focused on increasing muscle strength, rather than $k_T$ [55].

Opportunities remain to simultaneously address the metabolic consequences of both reduced $k_T$ and diminished $F_P$. For example, ankle exoskeletons could be designed to augment

ankle joint stiffness and provide supplemental $F_P$ with potential to reduce the metabolic cost of walking [41,56]. In addition, exoskeletons that can provide chronic (i.e. employable over weeks to months) wearable powered assistance or resistance on-demand could be used to interleave phases of on-line gait retraining to improve volitional Fp with phases of scheduled resistance to increase muscle strength and tendon stiffness [57]. Ultimately, this stresses that holistic design, iterative functional testing, and personalized prescriptions may be needed to implement effective wearable devices or rehabilitative therapies to combat walking inefficiency and deteriorating functional ability to maintain overall health in our aging population.

## Limitations

First, it is very time consuming (weeks of plyometric training or immobilization) to conduct a study to alter $k_T$ in human subjects. Furthermore, prospective tendon overloading or underloading protocols cannot isolate adaptations only to the tendon as the associated musculature will also be affected. Thus, we relied on the combination of experimental procedures and targeted modeling approach to explore the metabolic cost-$k_T$-$F_P$ landscape. A noted by others [58], simulations cannot exactly replicate the motions and forces a participant may produce at both the specified $k_T$ and $F_P$. Thus, our experiments may not offer exactly the appropriate constraints for estimating changes in muscle dynamics of actual participants with altered $k_T$. We only changed tendon strain ($\varepsilon$) in this study in attempt to address the independent effects of $k_T$. Yet, that single variable is unlikely to fully characterize age-related changes in musculotendon dynamics. Future studies may also alter other tendon properties within the Thelen2003 muscle model (such as tendon slack length or MTU passive shape factor) or utilize other musculotendon models that have a higher resolution for adjusting tendon parameters (i.e., Millard2012).

Second, our discoveries are, by design, inferred from data-driven musculoskeletal simulations rather than via direct empirical measures. Our simulations reported high and potentially implausible metabolic costs at the beginning of the gait cycle, relatively early in each simulation's run time. We suspect these outcomes are simulation artifacts rather than reliable predictions. Activation simulations must compute muscle state initial values (length, force, activation) when simulating a movement. Although we applied our simulation 0.05 s before the start of each gait cycle (larger than the recommended 0.03 s), muscle states may not have fully reached equilibrium, particularly at low stiffnesses. For this reason, we opted not to discuss or interpret any findings during very early stance phase (<5% gait cycle). Additionally, like most forward dynamic simulations, our musculoskeletal dynamics and metabolic cost outcomes depend on optimization algorithms with objective functions that seek to minimize the muscle activation squared. Although these simulation methods are supported by experimental evidence [29,59], human neuromechanics may not always align with their outcomes.

Third, we designed our biofeedback paradigm to prompt changes in $F_P$ while at a constant speed, which also resulted in changes in stride length [8]. We did not characterize the effects of stride length specifically in this study. However, because older adults exhibit reduced stride length [60] as well as $F_P$, we contend that the outcomes reported here are relevant to our target population. We have previously quantified and discussed at length these interactions between $F_P$ and stride length and their potential impact on metabolic cost [17,54].

## Conclusion

We combined computational and experimental analyses to answer the following question with clinically important implications: "Does walking with reduced $F_P$ mitigate the metabolic penalty of reduced $k_T$?" In an experimental paradigm designed for young adults to emulate older

adult walking via targeted $F_P$ biofeedback across various simulated $k_T$ levels, the answer is "no". Walking metabolic cost is elevated both with reduced $k_T$ or with any deviation in $F_P$ and we observed no trade-off that could enable functional adaptations to overcome altered structural properties of the musculoskeletal system. However, even though total metabolic cost increased by 5% on average with moderate reductions in $k_T$ and $F_P$, the triceps surae muscles did experience 7% local cost reductions on average. Although $k_T$ and $F_P$ may need to be quantified individually, they need not be addressed separately, as wearable devices and rehabilitative strategies could be designed to simultaneously address one or both these key factors driving metabolic cost.

## Supporting information

**S1 Fig. Muscle average activation levels.** We show individual-muscle activation levels as a function of both $k_T$ and $F_P$. ANOVA main effects are shown via arrows (horizontal, vertical, & diagonal) similar to Figs 2 & 4. Eleven of the 12 highest energy consuming muscles showed significant effects in activation level for $k_T$ (all but *glut_med*, panel D). Five of the 12 most costly muscles showed significant effects in activation level for $F_P$. Three out of the 12 muscles displayed significant interactions between $k_T$ and $F_P$ for activation level. These data correspond with S1 Table.
(TIF)

**S2 Fig. Muscle average fiber lengths.** We show individual-muscle fiber lengths as a function of both $k_T$ and $F_P$. The fiber lengths of all 12 of the costliest muscles were significantly impacted by $k_T$. Additionally, 8 of these top 12 had significant effects for $F_P$, while 7/12 showed significant interaction effects. These data correspond with S2 Table.
(TIF)

**S3 Fig. Muscle instantaneous activation levels.** The instantaneous activation levels for highest consuming muscles highly aligned with the instantaneous metabolic costs (Fig 5).
(TIF)

**S4 Fig. Muscle instantaneous fiber lengths.** Viewing the instantaneous muscle fiber lengths, we see changes across the experimental conditions for both $k_T$ and $F_P$. These showcase underlying changes in muscle actions as a result from the altered $k_T$ and $F_P$. Large effects occur in the distal musculature, particularly for the *solues*, *med_gas*, and *tib_ant* (J, K, & L).
(TIF)

**S5 Fig. Triceps surae metabolic costs.** Due to their large influence on ankle moment and thus $F_P$, we highlight the triceps surae metabolic cost on average (A-C) and across the gait cycle (D-F). One can see a high similarity between *med_gas* and *lat_gas* across $F_P$ and $k_T$ experimental conditions. This view of all three triceps surae muscles is inaccessible in other figures due to the small relative metabolic cost of *lat_gas*.
(TIF)

**S6 Fig. Lower body kinematics.** We show the influence of the $F_P$ biofeedback conditions on sagittal plane kinematics for the hip (A), knee (B), and ankle (C). Subject-averaged curves are surrounded by shading of ±1 standard deviation. We also calculated one-way repeated measures analysis of variance on the influence of $F_P$ condition on joint angles across the gait cycle using SPM. Like our other figures, a black bar at the top of each panel denotes a significant main effect for that instance of the gait cycle (1% increments). In general, reducing $F_P$ resulted in smaller dynamic ranges across all lower body joints.
(TIF)

**S7 Fig. Triceps surae impact of tendon stiffness on muscle dynamics.** To show the effects of $k_T$ on muscle length more clearly, this figure focusses on the *soleus* (A) and *med_gas* (B) muscles for the Norm $F_P$ condition. Both subplots show the average relative muscle fiber length with shaded ±1 standard deviation across the gait cycle for the least stiff (8% $\varepsilon_o$, blue solid line) and most stiff (2% $\varepsilon_o$, red dashed line) conditions. We remind readers that kinematics (and thus MTU lengths) are constrained for all simulations. Comparing the two $k_T$ extrema, we see a clear reduction in operating range for the 8% $\varepsilon_o$ condition across both muscles, implying that the more compliant tendon would elongate and recoil more in contribution to MTU length changes. Conversely, a stiffer tendon is unlikely to experience as much length change, requiring its respective muscle to undergo more shortening and lengthening. Indeed, in the least-stiff condition, the *med_gas* (B) acts nearly isometrically during mid-stance (20–50% gait cycle).
(TIF)

**S1 Table. We show the average activation level for all modeled muscles, averaged across the gait cycle.** We display how these individual muscles respond to changes in $F_P$, $k_T$, and interaction by reporting the ANOVA main effect (p-value) and effect size ($\eta_p^2$). Bolded muscle names indicate the top 12 consumers of metabolic cost, highlighted in Figs 4 & 5.
(DOCX)

**S2 Table. We show the average normalized fiber length for all modeled muscles, averaged across the gait cycle.** We display how these individual muscles respond to changes in $F_P$, $k_T$, and interaction by reporting the ANOVA main effect (p-value) and effect size ($\eta_p^2$). Bolded muscle names indicate the top 12 consumers of metabolic cost, highlighted in Figs 4 & 5.
(DOCX)

## Acknowledgments

We would like to thank the University of North Carolina at Chapel Hill and the Research Computing group for providing computational resources and support that have contributed to these research results. We acknowledge Noah Pieper, Sidney Baudendistal, Gabriela Diaz, and Rebecca Krupenevich for collecting the data that enabled this simulation study.

## Author Contributions

**Conceptualization:** Richard E. Pimentel, Jason R. Franz.

**Data curation:** Richard E. Pimentel.

**Formal analysis:** Richard E. Pimentel.

**Funding acquisition:** Gregory S. Sawicki, Jason R. Franz.

**Investigation:** Richard E. Pimentel.

**Methodology:** Richard E. Pimentel.

**Software:** Richard E. Pimentel.

**Supervision:** Gregory S. Sawicki, Jason R. Franz.

**Validation:** Richard E. Pimentel.

**Visualization:** Richard E. Pimentel.

**Writing – original draft:** Richard E. Pimentel.

**Writing – review & editing:** Richard E. Pimentel, Gregory S. Sawicki, Jason R. Franz.

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
