## [Decision Letter · Decision Letter 0]

15 Jun 2023

PONE-D-23-13303Simulations suggest walking with reduced propulsive force would not mitigate the energetic consequences of lower tendon stiffnessPLOS ONE

Dear Dr. Franz,

Thank you for submitting your manuscript to PLOS ONE. After careful consideration, we feel that it has merit but does not fully meet PLOS ONE’s publication criteria as it currently stands. That said, you will see that both reviewers were positive about the paper, and some of the comments are suggestions. Therefore, we invite you to submit a revised version of the manuscript that addresses the points raised during the review process.

We look forward to receiving your revised manuscript.

Kind regards,

Charlie M. Waugh

Academic Editor

PLOS ONE

“We thank our participants for volunteering their time and acknowledge the NIH for funding this work (R01AG058615). We would like to thank the University of North Carolina at Chapel Hill and the Research Computing group for providing computational resources and support that have contributed to these research results. We acknowledge Noah Pieper, Sidney Baudendistal, Gabriela Diaz, and Rebecca Krupenevich for collecting the data that enabled this simulation study.”

“JRF and GSS received funding for this study from National Institutes of Health (https://www.nih.gov/) Grant: R01AG058615. The funders had no role in study design, data collection and analysis, decision to publish, or preparation of the manuscript.”

3. Please include your tables as part of your main manuscript and remove the individual files. Please note that supplementary tables (should remain/ be uploaded) as separate "supporting information" files"

4. We note that Figure 1 in your submission contain copyrighted images. All PLOS content is published under the Creative Commons Attribution License (CC BY 4.0), which means that the manuscript, images, and Supporting Information files will be freely available online, and any third party is permitted to access, download, copy, distribute, and use these materials in any way, even commercially, with proper attribution. For more information, see our copyright guidelines: http://journals.plos.org/plosone/s/licenses-and-copyright.

Reviewers' comments:

Reviewer's Responses to Questions

**Comments to the Author**

1. Is the manuscript technically sound, and do the data support the conclusions?

Reviewer #1: Yes

Reviewer #2: Yes

2. Has the statistical analysis been performed appropriately and rigorously? 

Reviewer #1: Yes

Reviewer #2: Yes

3. Have the authors made all data underlying the findings in their manuscript fully available?

Reviewer #1: Yes

Reviewer #2: Yes

4. Is the manuscript presented in an intelligible fashion and written in standard English?

Reviewer #1: Yes

Reviewer #2: Yes

5. Review Comments to the Author

Reviewer #1: General questions:

The study engendered a sense of interest as a result of its well-crafted composition and effectively presented findings. The entirety of the methodological approach and the resultant outcomes are deemed satisfactory. Nonetheless, I have compiled a series of general and specific comments aimed at enhancing the manuscript's overall quality. The primary objective of the investigation revolves around ascertaining potential correlations between diminished horizontal ground reaction force exerted on the foot and decreased tendon stiffness to decrease metabolic cost of walking. Through the examination of this hypothesis, the authors meticulously scrutinized young, healthy individuals and endeavored to extrapolate their findings to the elderly population by employing musculoskeletal simulations.

The study's purpose and hypothesis were designed to answer a very specific question: "Could walking with a reduced Fp mitigate the metabolic penalty we pay for reduced kt?".

However, the method and results section includes extra information unrelated to the core research question. Notably, the mention of increased Fp beyond normal levels (i.e., +20% and +40%) lacks prior discussion in the introduction and abruptly emerges in the subsequent method and results section without appropriate contextualization for the reader. Consequently, I strongly urge the author to either provide a rationale and hypothesis for investigating the purpose of the study in relation to the increased Fp or consider removing the increased Fp from the study.

All participants initially determined their preferred walking speed during overground walking, which was subsequently employed as their default speed on the treadmill (referred to as Fp normal). Subsequently, the biofeedback system was utilized to modify the Fp level, adjusting it to ±20% and ±40% of the Fp normal value. While it is true that the walking speed remained constant (corresponding to the individuals' preferred walking speed), it is important to investigate whether the kinematic parameters, such as ankle, knee, and hip joint angles, retained the same range of motion.

It is widely recognized that even minor ankle joint rotations can substantially influence maximal tendon strain (Arampatzis et al., 2008). Moreover, significant differences in the joint range of motion for the lower extremities have been observed between young and elderly individuals (Fukuchi & Duarte, 2008). Thus, it is imperative to provide further clarification regarding this apparent discrepancy and its implications on the study's findings.

The authors answered the main question of the study (i.e., Could walking with a reduced Fp mitigate the metabolic penalty we pay for reduced kt?) to a solid "No". I think it can increase the impact and strength of the manuscript if the authors indicate which muscle-tendon (or group of muscle-tendon such as triceps-surae and Achilles tendon) can mitigate the increased metabolic cost of the muscle-tendon (muscle-tendon group) due to reduced kt by diminished Fp. For instance, the GM muscle in -20 % seems to be the answer.

Specific questions:

Line 130: I find it challenging to comprehend the rationale behind the utilization of the metabolic energy expenditure system in your study, considering that neither its role as a validation tool for the model (as previously conducted) nor its output data are intended to be reported.

Line 142: I would appreciate a more detailed explanation regarding the origin or basis of the 2.0% value for the default maximum tendon strain in the gait 2392 model, considering that the default value for all muscles in the model is fixed at 3.3%.

Figure 3: As a suggestion: Put the SPM significance bars on the bottom of the figure. In this way, the significant difference in the percentage of the gait cycle is more readable (it can be applied to other figures too). Or put the gait cycle ticks (5 ticks) on top of the SPM bar (I would prefer this one).

Figure 4: As a suggestion: I would like to see how the GL muscle reacts to the deviated Fp from normal and reduced Kt of the Achilles tendon (I expect to see something similar to GM). I know you have prioritized the muscle based on their metabolic expenditure, but the results of GL as one of the triceps surae muscle components can be interesting.

Arampatzis, A., De Monte, G., & Karamanidis, K. (2008). Effect of joint rotation correction when measuring the elongation of the gastrocnemius medialis tendon and aponeurosis. Journal of Electromyography and Kinesiology, 18(3), 503-508.

Fukuchi, R. K., & Duarte, M. (2008). Comparison of three-dimensional lower extremity running kinematics of young adult and elderly runners. Journal of sports sciences, 26(13), 1447-1454.

Reviewer #2: The present paper provides a comprehensive examination of the influence of tendon stiffness and push-off propulsive force on the energy cost of walking, two significant aging-related factors known to contribute to increased metabolic cost. The study employs an integrative experimental and computational approach, utilizing personalized musculoskeletal models and experimental data from a previous study in which 12 young adults were instructed to walk on a force-sensing treadmill while adjusting their push-off propulsive force based on biofeedback. Overall, the manuscript presents a well-executed original study that advances the understanding of aging-related changes in walking and their impact on metabolic energy cost, which is crucial for effectively targeting preventative and rehabilitative strategies for the elderly population. The manuscript is well structured, and its conclusions are supported by the data. The analyses followed a high technical standard and were described in sufficient detail.

I have a few minor comments that the author might consider before publication.

Lines 136–142: Although the model used has been described in a previous publication, please consider providing further specifications, including details about the built-in Thelen2003Muscle model, properties of the muscle-tendon units, and the performance criterion used to solve muscle redundancy. Including these details within the manuscript would enhance its accessibility and ensure a comprehensive understanding of the study.

Lines 142–143/267–268: The modeling approach employed in this study is commendable and contributes to the understanding of the investigated factors. To enhance the clarity and interpretation of the findings, it would be valuable to clarify whether the model keeps joint angles and joint moments the same within a push-off propulsive force condition or if they can vary. Additionally, briefly commenting on the potential effects of a greater elongation for a compliant tendon on range of motion or muscle-tendon unit length may reasonably complement the discussion.

Lines 261–297:

(i) Consider discussing potential adaptations of optimal fascicle length in older individuals (e.g., Stenroth et al., 2012, https://doi.org/10.1152/japplphysiol.00782.2012; Delabastita et al., 2020, https://doi.org/10.1101/2020.02.10.941591) and their impact on muscular efficiency during walking (e.g., Lichtwark and Wilson, 2008, https://doi.org/10.1016/j.jtbi.2008.01.018).

(ii) Although the force–velocity relationship was not directly investigated, it would be valuable to acknowledge its potential implications for the observed findings. Addressing how the force–length relationship and the force–velocity relationship may interact with tendon stiffness and push-off propulsive force could provide additional insights on the underlying mechanisms.

Lines 283–287: For clarity, consider providing numerical support for fascicle lengthening (e.g., lengthening amplitude within the 10–50% interval for both the least and most stiff tendons for the Fp-norm). Also, consider discussing any potential energy loss during these eccentric contractions.

Lines 311-312: The thorough discussion of the impact of tendon stiffness and push-off propulsive force on the energy cost of walking is appreciated. However, it is surprising that the interaction between these factors was not addressed in more detail. Although no interaction effect was observed for the total energy cost of walking, it is noteworthy that five of the 12 most energy-consuming muscles exhibited an interaction effect, particularly the three muscles spanning the ankle. Further discussion of these findings could provide valuable insights into specific muscle contributions and their potential implications for walking efficiency.

Line 332:

The term “unethical” is subjective. Instead, I recommend highlighting the advantages of the targeted modeling approach, which allows for the adaptation of individual structures with a high degree of specificity. Previous studies using interventions to adjust tendon stiffness, such as immobilization or strength training, could not isolate the effects on the tendon alone.

6. PLOS authors have the option to publish the peer review history of their article (what does this mean?). If published, this will include your full peer review and any attached files.

Reviewer #1: **Yes: **Mohamadreza Kharazi

Reviewer #2: No

---

## [Author Response · Author response to Decision Letter 0]

8 Sep 2023

[Please see our response to reviewers document for a clearer response, including text styling and figures]

Reviewer 1:

The study engendered a sense of interest as a result of its well-crafted composition and effectively presented findings. The entirety of the methodological approach and the resultant outcomes are deemed satisfactory. Nonetheless, I have compiled a series of general and specific comments aimed at enhancing the manuscript's overall quality. The primary objective of the investigation revolves around ascertaining potential correlations between diminished horizontal ground reaction force exerted on the foot and decreased tendon stiffness to decrease metabolic cost of walking. Through the examination of this hypothesis, the authors meticulously scrutinized young, healthy individuals and endeavored to extrapolate their findings to the elderly population by employing musculoskeletal simulations. 

We appreciate the thoughtful review and insight into this manuscript. We have adopted the reviewers suggestions as outlined in our responses that follow.

The study's purpose and hypothesis were designed to answer a very specific question: "Could walking with a reduced Fp mitigate the metabolic penalty we pay for reduced kt?". However, the method and results section includes extra information unrelated to the core research question. Notably, the mention of increased Fp beyond normal levels (i.e., +20% and +40%) lacks prior discussion in the introduction and abruptly emerges in the subsequent method and results section without appropriate contextualization for the reader. Consequently, I strongly urge the author to either provide a rationale and hypothesis for investigating the purpose of the study in relation to the increased Fp or consider removing the increased Fp from the study.

We have added the following rationale to our revised introduction to better justify our inclusion of increased FP conditions (lines 88-93):

Our central motivation was to determine whether walking with reduced FP mitigates the metabolic penalty of reduced kT. However, examining larger than usual FP values is an important scientific contribution, allowing for a more comprehensive understanding of the relation between FP, kT, and walking metabolic cost. Understanding the full landscape of both decreasing and increasing FP in response to biofeedback can enhance the clinical impact toward therapeutic interventions designed to enhance FP. 

All participants initially determined their preferred walking speed during overground walking, which was subsequently employed as their default speed on the treadmill (referred to as Fp normal). Subsequently, the biofeedback system was utilized to modify the Fp level, adjusting it to ±20% and ±40% of the Fp normal value. While it is true that the walking speed remained constant (corresponding to the individuals' preferred walking speed), it is important to investigate whether the kinematic parameters, such as ankle, knee, and hip joint angles, retained the same range of motion.

It is widely recognized that even minor ankle joint rotations can substantially influence maximal tendon strain (Arampatzis et al., 2008). Moreover, significant differences in the joint range of motion for the lower extremities have been observed between young and elderly individuals (Fukuchi & Duarte, 2008). Thus, it is imperative to provide further clarification regarding this apparent discrepancy and its implications on the study's findings.

We agree that this further clarification is important. We address these concerns regarding kinematic parameters by adding Supplementary Figure 6, which contains group-averaged sagittal plane kinematics across our experimental conditions. We agree that kinematic patterns can influence maximal tendon strain, and further clarify these points in the results on lines 260-262:

In terms of kinematics, we found that reducing FP tended to decrease sagittal plane hip, knee, and ankle joint ranges of motion (Supp. Fig. 6). Conversely, increasing FP tended to increase ankle extension near push-off (~60% gait cycle).

And our revised discussion on lines 358-364:

Our protocol prompted changes in FP using targeted biofeedback which changed lower body kinematics (Supplementary Fig. 6). Because even minor changes in joint kinematics can influence tendon length (and therefore strain)(51), we would infer the inverse is also true (where minor changes in tendon strain can influence joint kinematics). This inverse relationship is supported in part by differing joint kinematics between younger and older adults during walking. (52). Because there is no way to control for all experimental variables (kT, FP, simulated metabolic cost, and effects of aging) in this study, we note that joint kinematics did change in response to FP, but were constrained across the kT simulations.

Supplementary Figure 6: We show the influence of the FP biofeedback conditions on sagittal plane kinematics for the hip (A), knee (B), and ankle (C). Subject-averaged curves are surrounded by shading of ±1 standard deviation. We also calculated one-way repeated measures analysis of variance on the influence of FP condition on joint angles across the gait cycle using SPM. Like our other figures, a black bar at the top of each panel denotes a significant main effect for that instance of the gait cycle (1% increments). In general, reducing FP resulted in smaller dynamic ranges across all lower body joints. 

The authors answered the main question of the study (i.e., Could walking with a reduced Fp mitigate the metabolic penalty we pay for reduced kt?) to a solid "No". I think it can increase the impact and strength of the manuscript if the authors indicate which muscle-tendon (or group of muscle-tendon such as triceps-surae and Achilles tendon) can mitigate the increased metabolic cost of the muscle-tendon (muscle-tendon group) due to reduced kt by diminished Fp. For instance, the GM muscle in -20 % seems to be the answer.

We are grateful for the reviewer’s insight that med_gas (and lat_gas, albeit to a lesser extent) demonstrate the energy mitigation strategy to decrease FP in the face of lesser kT. We have clarified this point in our revise discussion and added this to one of our primary conclusions. 

Discussion lines 270-279: 

Our second hypothesis was more nuanced. We reject our second hypothesis at the whole-body level, as total metabolic cost only increased as kT and FP decreased (rightward and downward shift in Fig. 2). However, certain muscles did display cost savings when emulating age-related changes (lower kT and reduced FP). In particular, the triceps surae muscles are a primary determinant of FP generation and exhibited a rightward and downward shift (Supplementary Fig. 5A-C). For example, if we compare the default condition (Norm FP and 3.3% εo) with a reasonable aging shift (i.e., -20% FP and 6% εo) we see a 4.6% increase in total metabolic cost (Fig. 2) as well as a 7% cost reduction for the triceps surae (4.4%, 6.8%, and 9.9% reductions for the soleus, med_gas, and lat_gas, respectively, Fig. 4 & Supplementary Fig. 5). Thus, these simulations suggest that older adults may adopt a strategy with local reductions in metabolic cost, but with increased costs when summed at the whole-body level. 

Discussion lines 398-403: 

The triceps surae musculature (soleus, med_gas, and lat_gas) provides a significant portion of the work and power for gait by generating FP. Neuromuscular adaptations conserve energy on an individual-muscle or muscle-group level. We found that local neuromuscular responses to minimize activation (i.e., reducing FP in response to biofeedback) may successfully mitigate costs to operate the triceps surae muscles (Supplementary Fig. 5), but do not necessarily reduce total energy costs (Fig. 2) due to compensatory neuromuscular responses in other muscles (Fig. 4).

Specific questions: 

Line 130: I find it challenging to comprehend the rationale behind the utilization of the metabolic energy expenditure system in your study, considering that neither its role as a validation tool for the model (as previously conducted) nor its output data are intended to be reported.

We have removed this paragraph from the revised manuscript. 

Line 142: I would appreciate a more detailed explanation regarding the origin or basis of the 2.0% value for the default maximum tendon strain in the gait 2392 model, considering that the default value for all muscles in the model is fixed at 3.3%.

We have added a sentence to our revised methods section to cover this point, which now reads (lines 145-147): 

We chose these strain values because they lie within previous simulation studies (6,30), contain the expected range of tendon strain for younger and older adults (11,31,32), and, in the case of 2% εo, provides a stiffer comparison versus default (3.3% εo). 

Figure 3: As a suggestion: Put the SPM significance bars on the bottom of the figure. In this way, the significant difference in the percentage of the gait cycle is more readable (it can be applied to other figures too). Or put the gait cycle ticks (5 ticks) on top of the SPM bar (I would prefer this one). 

Great suggestion. We have added gait cycle ticks on top of the SPM bar in Figure 3 as recommended. 

Figure 4: As a suggestion: I would like to see how the GL muscle reacts to the deviated Fp from normal and reduced Kt of the Achilles tendon (I expect to see something similar to GM). I know you have prioritized the muscle based on their metabolic expenditure, but the results of GL as one of the triceps surae muscle components can be interesting.

Thank you for this feedback. Due to this request, we have added one additional supplementary figure that compares the 3 triceps surae muscles in more detail (Supplementary Figure 5 – Triceps Surae) and reference this figure in our revised results section on lines 256-260:

We conclude our results by providing additional data on the triceps surae musculature (Supplementary Fig. 5) and on lower-body sagittal plane kinematics across the biofeedback conditions (Supplementary Fig. 6). Supplementary Figure 5 summarizes comparisons between the three triceps surae muscles, as lat_gas was not a member of the top 12 contributors to metabolic cost. The cost landscape for lat_gas largely aligned with that of med_gas but with smaller metabolic costs.

The discussion on lines 272-279: 

However, certain muscles did display cost savings when emulating age-related changes (lower kT and reduced FP). In particular, the triceps surae muscles are a primary determinant of FP generation and exhibited a rightward and downward shift (Supplementary Fig. 5A-C). For example, if we compare the default condition (Norm FP and 3.3% εo) with a reasonable aging shift (i.e., -20% FP and 6% εo) we see a 4.6% increase in total metabolic cost (Fig. 2) as well as a 7% cost reduction for the triceps surae (4.4%, 6.8%, and 9.9% reductions for the soleus, med_gas, and lat_gas, respectively, Fig. 4 & Supplementary Fig. 5). Thus, these simulations suggest that older adults may adopt a strategy with local reductions in metabolic cost, but with increased costs when summed at the whole-body level. 

Supplementary Figure 5: Due to their large influence on ankle moment and thus FP, we highlight the triceps surae metabolic cost on average (A-C) and across the gait cycle (D-F). One can see a high similarity between med_gas and lat_gas across FP and kT experimental conditions. This view of all three triceps surae muscles is inaccessible in other figures due to the small relative metabolic cost of lat_gas.

Reviewer 2:

The present paper provides a comprehensive examination of the influence of tendon stiffness and push-off propulsive force on the energy cost of walking, two significant aging-related factors known to contribute to increased metabolic cost. The study employs an integrative experimental and computational approach, utilizing personalized musculoskeletal models and experimental data from a previous study in which 12 young adults were instructed to walk on a force-sensing treadmill while adjusting their push-off propulsive force based on biofeedback. Overall, the manuscript presents a well-executed original study that advances the understanding of aging-related changes in walking and their impact on metabolic energy cost, which is crucial for effectively targeting preventative and rehabilitative strategies for the elderly population. The manuscript is well structured, and its conclusions are supported by the data. The analyses followed a high technical standard and were described in sufficient detail. I have a few minor comments that the author might consider before publication.

We appreciate the reviewer’s constructive comments and suggestions. We have adopted these recommendations into the revised version of our manuscript as outlined in our responses that follow.

Lines 136–142: Although the model used has been described in a previous publication, please consider providing further specifications, including details about the built-in Thelen2003Muscle model, properties of the muscle-tendon units, and the performance criterion used to solve muscle redundancy. Including these details within the manuscript would enhance its accessibility and ensure a comprehensive understanding of the study.

Thank you for reminding us to elaborate on these important model details. We have updated our model description in the methods to now read: (lines 134-140)

These musculoskeletal models use Hill-type muscle models (Thelen2003Muscle) with standard equilibrium equations to simulate musculotendon dynamics (26). Model parameters based on anthropometrics (optimal fiber length and tendon slack length) were scaled using segmental weighting factors. Similar to previous modeling studies (27,28), we scaled maximum isometric force by 1.5 times default value to ensure all simulations could produce the requisite joint moments. We maintained defaults for all other model parameters. We also maintained the classic optimization function that minimizes the square of muscle activations when computing muscle dynamics.

We have also added this in the limitations lines 432-436: 

We only changed tendon strain (ε) in this study in attempt to address the independenteffects of kT. Yet, that single variable is unlikely to fully characterize age-related changes in musculotendon dynamics. Future studies may also alter other tendon properties within the Thelen2003 muscle model (such as tendon slack length or MTU passive shape factor) or utilize other musculotendon models that have a higher resolution for adjusting tendon parameters (i.e., Millard2012).

Lines 142–143/267–268: The modeling approach employed in this study is commendable and contributes to the understanding of the investigated factors. To enhance the clarity and interpretation of the findings, it would be valuable to clarify whether the model keeps joint angles and joint moments the same within a push-off propulsive force condition or if they can vary. Additionally, briefly commenting on the potential effects of a greater elongation for a compliant tendon on range of motion or muscle-tendon unit length may reasonably complement the discussion.

We appreciate the comments of both reviewers on providing additional context for reported changes. Although we opted to omit joint moments from our reporting, we have revised our manuscript and supplementary material to include a comprehensive comparison of lower extremity joint kinematics. We copy our response to Reviewer 1 here for convenience.

We agree that this further clarification is important. We address these concerns regarding kinematic parameters by adding Supplementary Figure 6, which contains group-averaged sagittal plane kinematics across our experimental conditions. We agree that kinematic patterns can influence maximal tendon strain, and further clarify these points in the results on lines 260-262:

In terms of kinematics, we found that reducing FP tended to decrease sagittal plane hip, knee, and ankle joint ranges of motion (Supp. Fig. 6). Conversely, increasing FP tended to increase ankle extension near push-off (~60% gait cycle).

And our revised discussion on lines 358-364:

Our protocol prompted changes in FP using targeted biofeedback which changed lower body kinematics (Supplementary Fig. 6). Because even minor changes in joint kinematics can influence tendon length (and therefore strain)(51), we would infer the inverse is also true (where minor changes in tendon strain can influence joint kinematics). This inverse relationship is supported in part by differing joint kinematics between younger and older adults during walking.(52) Because there is no way to control for all experimental variables (kT, FP, simulated metabolic cost, and effects of aging) in this study, we note that joint kinematics did change in response to FP, but were constrained across the kT simulations.

[See Supplementary Figure 6] 

In addition, we have taken the reviewer’s recommendation to elaborated on the elongation effects of compliant tendons in our revised discussion on lines 295-298: 

Our main premise is that operating a less-stiff tendon elicits shorter muscle lengths to compensate for greater tendon elongation for a given force output and MTU length (6). These effects presume higher excitations at a metabolic penalty or may sufficiently alter musculotendon dynamics to alter muscle activation timing.

Lines 261–297:

(i) Consider discussing potential adaptations of optimal fascicle length in older individuals (e.g., Stenroth et al., 2012, https://doi.org/10.1152/japplphysiol.00782.2012; Delabastita et al., 2020, https://doi.org/10.1101/2020.02.10.941591) and their impact on muscular efficiency during walking (e.g., Lichtwark and Wilson, 2008, https://doi.org/10.1016/j.jtbi.2008.01.018).

We have added a paragraph on this (lines 299-304):

In a similar study to ours, increasing Achilles tendon stiffness did not significantly decrease walking energy cost in older adults (23). Biological ranges of kT incorporate the local cost minima during walking.(23,39) We generally interpret reduced kT as pathophysiological change that would benefit from intervention. However, as a potential alternative interpretation, older adults likely participate in less physical activity, and thus their lower stiffness may be a physiological adaptation to optimize metabolic cost for their daily activities (10).

(ii) Although the force–velocity relationship was not directly investigated, it would be valuable to acknowledge its potential implications for the observed findings. Addressing how the force–length relationship and the force–velocity relationship may interact with tendon stiffness and push-off propulsive force could provide additional insights on the underlying mechanisms.

Coming back to these F-V and F-L relationships and how kT and FP interact with them has been a useful thought experiment to understand underlying mechanisms. We have included the interaction between F-V and F-L relations in governing metabolic cost in our revised discussion on lines 333-339: 

The large effects of kT on metabolic cost may arise because it impacts not only the force-length, but also the force-velocity relation of muscle. For force-length, a less-stiff tendon would compel shorter muscle operating ranges when MTU length is constrained (as it was in this study). Shorter muscles generate less force, thus requiring a higher activation at a given submaximal force output, yielding higher metabolic costs. For force-velocity, a less-stiff tendon would elongate more when transmitting a specific force. In MTU-shortening activity, this would require the muscle to shorten more rapidly, requiring higher activations and thus, higher costs.

Lines 283–287: For clarity, consider providing numerical support for fascicle lengthening (e.g., lengthening amplitude within the 10–50% interval for both the least and most stiff tendons for the Fp-norm). Also, consider discussing any potential energy loss during these eccentric contractions.

As suggested, we have added a supplementary figure that more clearly shows this fiber lengthening, using the two extreme values of kT for comparison. Our revised manuscript references these data as follows (lines 323-327): 

If timed appropriately, tendons with lower stiffness may exploit tendon elongation to store spring potential energy for powering forceful activities (such as FP) later. Indeed, med_gas MTUs with the less-stiff tendons exhibit bursts of higher costs at the beginning of push-off (~40% gait cycle), but dramatic cost savings later in push-off (50-60% gait cycle, Fig. 5K). For a clearer distinction of these fiber length differences, please refer to Supplementary Fig. 7.

Supplementary Figure 7: To show the effects of kT on muscle length more clearly, this figure focusses on the soleus (A) and med_gas (B) muscles for the Norm FP condition. Both subplots show the average relative muscle fiber length with shaded ±1 standard deviation across the gait cycle for the least stiff (8% εo, blue solid line) and most stiff (2% εo, red dashed line) conditions. We remind readers that kinematics (and thus MTU lengths) are constrained for all simulations. Comparing the two kT extrema, we see a clear reduction in operating range for the 8% εo condition across both muscles, implying that the more compliant tendon would elongate and recoil more in contribution to MTU length changes. Conversely, a stiffer tendon is unlikely to experience as much length change, requiring its respective muscle to undergo more shortening and lengthening. Indeed, in the least-stiff condition, the med_gas (B) acts nearly isometrically during mid-stance (20-50% gait cycle).

Lines 311-312: The thorough discussion of the impact of tendon stiffness and push-off propulsive force on the energy cost of walking is appreciated. However, it is surprising that the interaction between these factors was not addressed in more detail. Although no interaction effect was observed for the total energy cost of walking, it is noteworthy that five of the 12 most energy-consuming muscles exhibited an interaction effect, particularly the three muscles spanning the ankle. Further discussion of these findings could provide valuable insights into specific muscle contributions and their potential implications for walking efficiency.

We first note that we discovered an error in the original Figure 3, which failed to show any significant instantaneous interaction effects. After regenerating our figures for these responses, we updated this figure as well as our results to reflect these differences. Specifically, we have revised our results section on lines 194-196 as follows: 

Although we found no interaction between kT and FP for total metabolic cost on average, we observed a few interactions during early stance and push-off (Fig. 3).

We have also added a new discussion section titled Interactions between Propulsive force and Tendon Stiffness. For convenience, this section now reads (lines 365-396): 

Although we did not find an interaction between kT and FP for total metabolic cost on average (Figure 2), we did see periods of significant interactions across the gait cycle (Figure 3) occurring during early stance (~5-20%) and push off (~55%). Because the instantaneous total metabolic costs in Figure 3 are comprised of bilateral data, the periods of significant interaction (5-10% and 55-60% gait cycle) may arise from the bilateral nature of gait, with both distal and proximal muscles contributing to the interactions across both time periods. As total metabolic cost is the sum from individual muscles, we discuss the individual muscles likely to explain these interactions. 

Of the 5 muscles with a significant kT and FP interaction on average (Figure 4), glut_max and vas_lat likely explain those during early stance. The cost profile for glut_max during early stance (5-10%) exhibited higher metabolic costs for both lower FP (major vertical axis) and lower kT (minor vertical axis, Fig. 5C). The higher costs due to lower FP aligns with the well-documented redistribution of muscle work from the ankle to the hip (16). The effect of kT likely arises from the relative fiber length differences imposed by tendon constraints. During this concentric activity of glut_max (Supplementary Fig. 4C), some of the MTU shortening is lost via tendon elongation, requiring glut_max to contract more to overcome the tendon lengthening, and thus requiring higher costs (Fig. 5C). 

In an eccentric example, vas_lat consumes more energy with higher kT (Fig. 5F) during eccentric activity during early-to-mid stance (15-20% gait cycle, Supplementary Fig. 3F, 4F). Vas_lat muscle fibers attached to less-stiff tendons (higher εo) may maintain relatively similar lengths as the compliant tendons uptake changes in length. Whereas, when attached to stiff tendons, vas_lat muscle fibers must elongate to account for the length change (see white/green areas in the minor axis of Supplementary Fig. 4F at 15-20% gait cycle, particularly visible for Norm and +40% FP conditions). This eccentric activity is more costly than isometric activity, exhibiting one aspect of the kT – FP interaction. 

Of the final 3 muscles that experienced significant interactions on average (Figure 4J,K,L), soleus and tib_ant demonstrated instantaneous interactions near the end of push-off (55-60% gait cycle, Fig. 5J,L). The soleus showed lower costs with reduced FP as well as lower costs with lower kT (higher εo). The lower metabolic costs with reduced FP is straightforward and aligns with lower soleus activation (Supplementary Fig. 3J). The lower metabolic cost at lower kT likely arises from elastic recoil of the Achille’s tendon providing some MTU-shortening this late into push-off. The interaction between kT and FP for the soleus also aligns with interaction timing for total metabolic cost (Fig. 3). Interestingly, med_gas did not share the instantaneous kT and FP interaction with the soleus during push-off, even though there were independent significant effects for kT and FP separately (Fig. 5). 

Line 332:

The term “unethical” is subjective. Instead, I recommend highlighting the advantages of the targeted modeling approach, which allows for the adaptation of individual structures with a high degree of specificity. Previous studies using interventions to adjust tendon stiffness, such as immobilization or strength training, could not isolate the effects on the tendon alone.

We appreciate this suggestion. We have rephrased this section to now read: (lines 425-428)

First, it is very time consuming (weeks of plyometric training or immobilization) to conduct a study to alter kT in human subjects. Furthermore, prospective tendon overloading or underloading protocols cannot isolate adaptations only to the tendon as the associated musculature will also be affected. Thus, we relied on the combination of experimental procedures and targeted modeling approach…

---

## [Decision Letter · Decision Letter 1]

11 Oct 2023

Simulations suggest walking with reduced propulsive force would not mitigate the energetic consequences of lower tendon stiffness

PONE-D-23-13303R1

Dear Dr. Franz,

We’re pleased to inform you that your manuscript has been judged scientifically suitable for publication and will be formally accepted for publication once it meets all outstanding technical requirements.

Please do review some additional (optional) suggestions from reviewer 2, in the event you would like to make final (minor) changes during the proof process.

Kind regards,

Charlie M. Waugh

Academic Editor

PLOS ONE

Additional Editor Comments (optional):

Reviewers' comments:

Reviewer's Responses to Questions

**Comments to the Author**

1. If the authors have adequately addressed your comments raised in a previous round of review and you feel that this manuscript is now acceptable for publication, you may indicate that here to bypass the “Comments to the Author” section, enter your conflict of interest statement in the “Confidential to Editor” section, and submit your "Accept" recommendation.

Reviewer #1: All comments have been addressed

Reviewer #2: All comments have been addressed

2. Is the manuscript technically sound, and do the data support the conclusions?

Reviewer #1: Yes

Reviewer #2: Yes

3. Has the statistical analysis been performed appropriately and rigorously? 

Reviewer #1: Yes

Reviewer #2: Yes

4. Have the authors made all data underlying the findings in their manuscript fully available?

Reviewer #1: Yes

Reviewer #2: Yes

5. Is the manuscript presented in an intelligible fashion and written in standard English?

Reviewer #1: Yes

Reviewer #2: Yes

6. Review Comments to the Author

Reviewer #1: The authors have thoroughly addressed all my questions and recommendations. I have no more suggestion to add.

Reviewer #2: Thank you for the thorough response and revisions. The addition of the informative supplementary figures has significantly strengthened the manuscript's quality. Before final publication, I have three minor aspects that might be worth addressing if considered relevant.

Lines 299-304: In the present model, it is clear that not all factors distinguishing older from younger can be considered. Nevertheless, a potential shorter optimal fascicle length in older people might be worth mentioning, as evidenced by Stenroth et al. (2012) and Delabastita et al. (2020). In fact, Delabastita et al. (2020) observed that shorter fascicles and a more compliant tendon had no effect on metabolic cost, suggesting that a shorter optimal fascicle length could be an adaptive response to offset increased tendon compliance in older adults.

Supplementary Figure 7: The near isometric behaviour of the muscle fascicle with the compliant tendon might be metabolically advantageous if optimal fascicle length is adapted, as the muscle not only operates close to its optimal length but also at a speed favourable for force generation while the energy is conserved in the tendon instead of being absorbed by the muscle during the eccentric phase. Related, Mian et al. 2007 (doi: 10.1111/j.1748-1716.2006.01634.x) observed differences in fascicle behaviour of the gastrocnemius lateralis between older and younger participants, which align well with the model's outcomes presented in Supplementary Figure 7.

Lines 359 - 362: Clarifying the cause and effect in the description linking joint kinematics to tendon length would enhance understanding, especially since the manuscript suggests force and intrinsic stiffness modulate tendon length.

7. PLOS authors have the option to publish the peer review history of their article (what does this mean?). If published, this will include your full peer review and any attached files.

Reviewer #1: **Yes: **Mohamadreza kharazi

Reviewer #2: No

---

## [Editor Report · Acceptance letter]

17 Oct 2023

PONE-D-23-13303R1 

Simulations suggest walking with reduced propulsive force would not mitigate the energetic consequences of lower tendon stiffness 

Dear Dr. Franz:

I'm pleased to inform you that your manuscript has been deemed suitable for publication in PLOS ONE. Congratulations! Your manuscript is now with our production department. 

Kind regards, 

on behalf of

Dr. Charlie M. Waugh 

Academic Editor

PLOS ONE